# Environmental response in gene expression and DNA methylation reveals factors influencing the adaptive potential of *Arabidopsis lyrata*

**Tuomas Hämälä\*‡, Weixuan Ning§, Helmi Kuittinen, Nader Aryamanesh†#, Outi Savolainen†**

Department of Ecology and Genetics, University of Oulu, Oulu, Finland

**\*For correspondence:** tuomas.hamala@gmail.com

†These authors contributed equally to this work

**Present address:** ‡School of Life Sciences, University of Nottingham, Nottingham, United Kingdom; §School of Natural Sciences, Massey University, Palmerston North, New Zealand; #Embryology Research Unit, Bioinformatics Group, Children's Medical Research Institute, University of Sydney, Westmead, Australia

**Competing interest:** The authors declare that no competing interests exist.

**Abstract** Understanding what factors influence plastic and genetic variation is valuable for predicting how organisms respond to changes in the selective environment. Here, using gene expression and DNA methylation as molecular phenotypes, we study environmentally induced variation among *Arabidopsis lyrata* plants grown at lowland and alpine field sites. Our results show that gene expression is highly plastic, as many more genes are differentially expressed between the field sites than between populations. These environmentally responsive genes evolve under strong selective constraint – the strength of purifying selection on the coding sequence is high, while the rate of adaptive evolution is low. We find, however, that positive selection on cis-regulatory variants has likely contributed to the maintenance of genetically variable environmental responses, but such variants segregate only between distantly related populations. In contrast to gene expression, DNA methylation at genic regions is largely insensitive to the environment, and plastic methylation changes are not associated with differential gene expression. Besides genes, we detect environmental effects at transposable elements (TEs): TEs at the high-altitude field site have higher expression and methylation levels, suggestive of a broad-scale TE activation. Compared to the lowland population, plants native to the alpine environment harbor an excess of recent TE insertions, and we observe that specific TE families are enriched within environmentally responsive genes. Our findings provide insight into selective forces shaping plastic and genetic variation. We also highlight how plastic responses at TEs can rapidly create novel heritable variation in stressful conditions.

## Editor's evaluation

This work provides a thorough look at changes in expression, methylation, and nucleotide and transposable element diversity among three populations of Arabidopsis lyrata in two different environments. It is a rich dataset, and the authors present a number of nice findings with relevance for our understanding of local adaptation and the process of – and potential constraints to – adaptation to rapid climate change.

## Introduction

To maintain viability in a changing environment, populations need to track shifting fitness optima through genetic adaptation or phenotypic plasticity (*Alberto et al., 2013*; *Ghalambor et al., 2007*; *Parmesan, 2006*). Although environmentally induced variation within a generation can efficiently facilitate population persistence in novel environments, selection on heritable variation is ultimately required for long-term adaptability (*Wright, 1931*). As such, strong phenotypic plasticity may constrain

adaptive evolution by masking genetic variation from directional selection (*Falconer, 1981*; *Grant, 1977*; *Wright, 1931*). On the other hand, increased population persistence due to plasticity might provide more opportunities for selection to act on heritable variation, making adaptive plasticity an important first step in evolution (*Baldwin, 1896*; *Schmalhausen, 1949*; *Waddington, 1942*). Given the importance of phenotypic plasticity in adaptive evolution, understanding the selective forces that shape plastic and genetic variation is valuable for predicting how organisms respond to both natural and human-mediated selection pressure.

In most cases, plasticity in morphological, developmental, and physiological traits is thought to result from changes in gene expression (*West-Eberhard, 2003*), making the study of expression plasticity a promising approach for uncovering the genetic basis of phenotypic plasticity. Although studies of environmentally responsive genes have discovered a wide range of expression responses (*Hodgins-Davis and Townsend, 2009*), many populations and species have reacted consistently to environmental stress (*He et al., 2021*; *Lowry et al., 2013*; *Wos et al., 2021a*; *Yeaman et al., 2014*). As gene expression is primarily controlled by regulatory elements acting either in cis (affecting nearby genes) or trans (affecting distant genes), the conserved expression responses are indicative of conserved regulatory systems (*Horvath et al., 2021*; *Lu et al., 2019*; *Rodgers-Melnick et al., 2016*). Such consistently expressed genes have also shown signals of strong purifying selection at the coding regions (*Hodgins et al., 2016*; *Hunt et al., 2013*; *Lasky et al., 2014*; *Lowry et al., 2013*; *Zhang et al., 2017*), suggesting that the regulatory conservation is frequently coupled with the conservation of the gene product. Consistent with this expectation, genes exhibiting genetically variable responses to the environment have harbored signals of relaxed selective constraint (*Hodgins et al., 2016*; *Koenig et al., 2013*; *Lasky et al., 2014*; *Zhang et al., 2017*). Given these general observations, we may expect that strong evolutionary conservation at environmentally responsive genes limits the emergence of heritable variation, whereas such variation is more likely to arise in rapidly evolving genes.

Besides inducing phenotypic plasticity through changes in gene expression, environmental stress may invoke plastic responses at transposable elements (TEs) (*Capy et al., 2000*; *Casacuberta and González, 2013*; *Ito et al., 2011*; *McClintock, 1984*; *Pietzenuk et al., 2016*; *Wos et al., 2021b*). TEs are commonly divided into two major classes depending on their mode of transposition: retrotransposons (or class I) that move by 'copy-and-paste' mechanism and DNA transposons (or class II) that move by 'cut-and-paste' mechanism (*Wicker et al., 2007*). Both classes can be further separated into distinct orders and superfamilies, which often occupy different 'niches' within the host genome (*Stitzer et al., 2021*). The activation of TEs leads both to proliferation of new TE copies and mobility of existing ones, which can have a considerable influence on the adaptive potential of a population (*Bourgeois and Boissinot, 2019*). On one hand, a large majority of new TEs and other structural variants are expected to be deleterious (*Baduel et al., 2021*; *Bourgeois and Boissinot, 2019*; *Hämälä et al., 2021*; *Kou et al., 2020*), and so TE activation likely increases the genetic load of a population. On the other hand, such activation can create novel functional variants, which may facilitate adaptation under new selective environments (*Capy et al., 2000*; *Casacuberta and González, 2013*; *Ito et al., 2011*; *McClintock, 1984*). Indeed, TEs have been associated with the emergence of several adaptive phenotypes, including industrial melanism in peppered moth (*Van't Hof et al., 2016*), early flowering in *Arabidopsis thaliana* (*Quadrana et al., 2016*), and single-stalk branching pattern in maize (*Studer et al., 2011*).

Here, we conducted a reciprocal transplant experiment to study selective processes underlying environmental responses. To do so, we grew *Arabidopsis lyrata* plants from three populations at natural low- and high-altitude field sites, leading to sharp differences in exposure to abiotic (e.g. temperature and solar radiation) and biotic (e.g. herbivores and pathogens) factors. Besides examining variation in gene expression, we searched for differentiation in DNA methylation between our experimental plants. DNA methylation is a common epigenetic modification that modulates gene expression (*Law and Jacobsen, 2010*). The regulatory mechanisms underlying DNA methylation can be rapidly activated by the environment, thus modifying gene expression in response to changing environmental conditions (*Liu and He, 2020*; *Thiebaut et al., 2019*). In plants, DNA methylation may also be transmitted from parent to offspring (*Law and Jacobsen, 2010*; *Lloyd and Lister, 2022*), although most stably inherited methylation variants have a genetic (and not plastic) basis (*Kawakatsu et al., 2016*; *Lloyd and Lister, 2022*; *Muyle et al., 2022*). DNA methylation is further involved in epigenetic silencing of TEs (*Law and Jacobsen, 2010*), and thus environmentally induced changes in

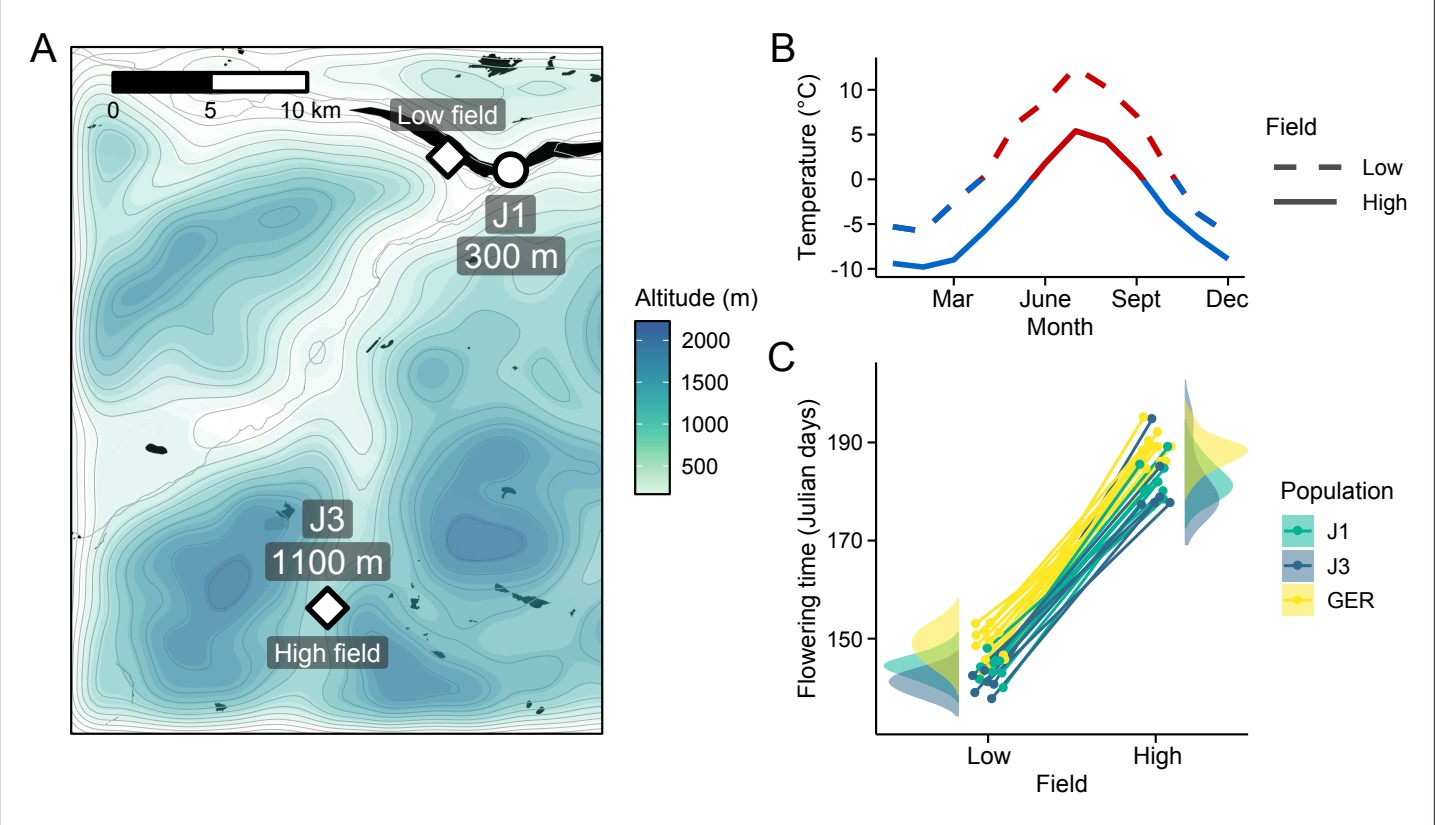

**Figure 1.** Reciprocal transplant experiment to study environmental adaptation in *A. lyrata*. (**A**) Locations and altitudes of the Norwegian populations and field sites. Map tiles by Stamen Design, under CC BY 3.0. Map data by OpenStreetMap, under ODbL. Altitude data from Shuttle Radar Topography Mission (SRTM) (*Farr and Kobrick, 2000*). (**B**) Average monthly temperature at areas around the field sites (red line: ≥0°C, blue line: <0°C). Data from MET Norway. (**C**) Average flowering time of full-sib families grown at the field sites. Data from *Hämälä et al., 2018*.

The online version of this article includes the following figure supplement(s) for figure 1:

**Figure supplement 1.** Locations of the *A. lyrata* populations.

**Figure supplement 2.** Photos of the low- and high-altitude field sites, taken during June 6 and 7, 2015, respectively.

the DNA methylome could either promote or suppress TE activity. Therefore, characterizing patterns of DNA methylation in naturally contrasting conditions may help us to better understand environmental effects in gene expression and TE activity. Using these data, we address the following questions: how are gene expression and DNA methylation influenced by the growing environment and population history? Is differential methylation associated with differential gene expression? Are differentially expressed and methylated genes under strong or relaxed evolutionary constraint? Do we find evidence of environmentally induced TE activation? Does TE activation create novel genetic variation capable of influencing environmental adaptation?

## Results

To study patterns of short- and long-distance local adaptation in *A. lyrata*, we grew plants from three populations at two contrasting common garden sites in Norway (*Figure 1A*). At both low- (300 meters above sea level [m.a.s.l.]) and high-altitude (1100 m.a.s.l.) field, we planted individuals from local Norwegian populations (J1, 300 m.a.s.l.; J3, 1100 m.a.s.l.) as well as individuals from a nonlocal population from Germany (GER; *Figure 1—figure supplement 1*). Although the field sites were closely situated (*Figure 1A*), the altitudinal difference resulted in considerable environmental differences between the fields (*Figure 1B* and *Figure 1—figure supplement 2*), thus leading to distinct responses in ecologically important traits (*Figure 1C*). Using multi-year fitness estimated during the experiment, we found evidence of home-site advantage between the J1 and J3 populations, while individuals

from the GER population fared poorly at both field sites (*Hämälä et al., 2018*). Here, we combine this reciprocal transplant experiment with transcriptome and whole-genome bisulfite sequencing (WGBS) to examine variation in gene expression and DNA methylation. Leaf samples from the plants were collected 1 year after planting, thus allowing long-term exposure to the environment. By comparing the field sites and populations, we can distinguish between three types of expression and methylation responses: (1) plastic (difference between the fields, but no difference between populations), (2) genetic (difference between populations, but no difference between the fields), and (3) plastic × genetic (difference between the fields and populations).

## Patterns of variation in gene expression and DNA methylation

We first used principal component analysis (PCA) to detect the main sources of variation in the gene expression and methylation data. In the gene expression data, the first principal component (PC) primarily corresponded to differences between the two field sites, whereas the Norwegian populations (J1 and J3) were differentiated from GER along the second PC axis (*Figure 2A*). By contrast, DNA methylation was mainly influenced by population structure, with first two PCs differentiating the three populations (*Figure 2A*). Separating DNA methylation into three sequence contexts, CG, CHG, and CHH (where H is A, T, or C), revealed that variation in CG methylation was congruent with the total methylation data (*Figure 2—figure supplement 1*). Methylation in the CHG and CHH contexts was primarily affected by differences between the Norwegian populations and GER, but the second PC in both PCAs revealed a slight influence of the field site, particularly in the Norwegian populations (*Figure 2—figure supplement 1*).

All three populations contained genes that were differentially expressed between the fields (*Figure 2B*), but the largest number of DEGs (differentially expressed genes) was found in J1. Expression differences between populations were mainly due to DEGs between the Norwegian populations and GER (*Figure 2C*), with a total of 1675 DEGs between GER and either J1 or J3. By contrast, we only detected 31 DEGs between J1 and J3 (*Figure 2C*). Therefore, environmental effects were evident in all populations, whereas population effects mainly arose from differences between Norway and GER.

Levels of DNA methylation were highly variable between the three contexts, with approximately 30, 10, and 3.5% of cytosines methylated in the CG, CHG, and CHH contexts, respectively (*Figure 2D*). CG methylation levels showed only a subtle difference between the field sites but a clear difference between the populations (*Figure 2D*); both J1 and J3 had considerably higher methylation levels than GER ($p < 2 \times 10^{-16}$, likelihood-ratio test [LRT]). For both CHG and CHH contexts, the Norwegian populations had higher methylation levels at the high-altitude field site ($p < 2 \times 10^{-16}$, LRT), whereas GER had higher methylation levels at the low-altitude field site ($p < 2 \times 10^{-16}$, LRT).

## Plastic and genetic responses at gene expression and DNA methylation

We found ample variation in gene expression and DNA methylation among our experimental plants. To more clearly distinguish the source of the variation, we used LRTs in DESeq2 (*Love et al., 2014*) to detect DEGs due to field site (DEG ~ field), population (DEG ~ population), and their interaction (DEG ~ field:population). We also used logistic regression and LRTs to conduct a similar analysis among differentially methylated genes (DMGs). As genic methylation can have distinct effects on gene expression depending on the sequence context (*Muyle et al., 2022*), we searched for DMGs using CG and non-CG methylation separately.

We detected between 112 and 3456 genes that had their expression or methylation levels affected by the field site and/or population history (*Table 1*). Consistent with the PCA results, most DEGs were found between the field sites (DEG ~ field), and most DMGs were found between populations (DMG ~ population). Despite the relative lack of methylation plasticity, differential methylation between the field sites was more common at non-CG than CG sites (*Table 1*). Field × population interactions (DEG ~ field:population) were rare in gene expression, and using the same criteria for outlier detection as with DEG ~ field and DEG ~ population, only 28 genes passed the DEG threshold. As this small number did not allow us to examine selective signals at the field × population genes, we used a more lenient threshold for multiple correction (see Materials and methods), while acknowledging the potentially higher false-positive rate among these genes. Of the 21,969 genes expressed in our experimental plants, we defined 3933 (18%) as environmentally responsive. Although methylation levels were correlated with gene expression levels (Spearman's $\rho_{CG} = 0.29$, $\rho_{CHG} = -0.23$, $\rho_{CHH} =$

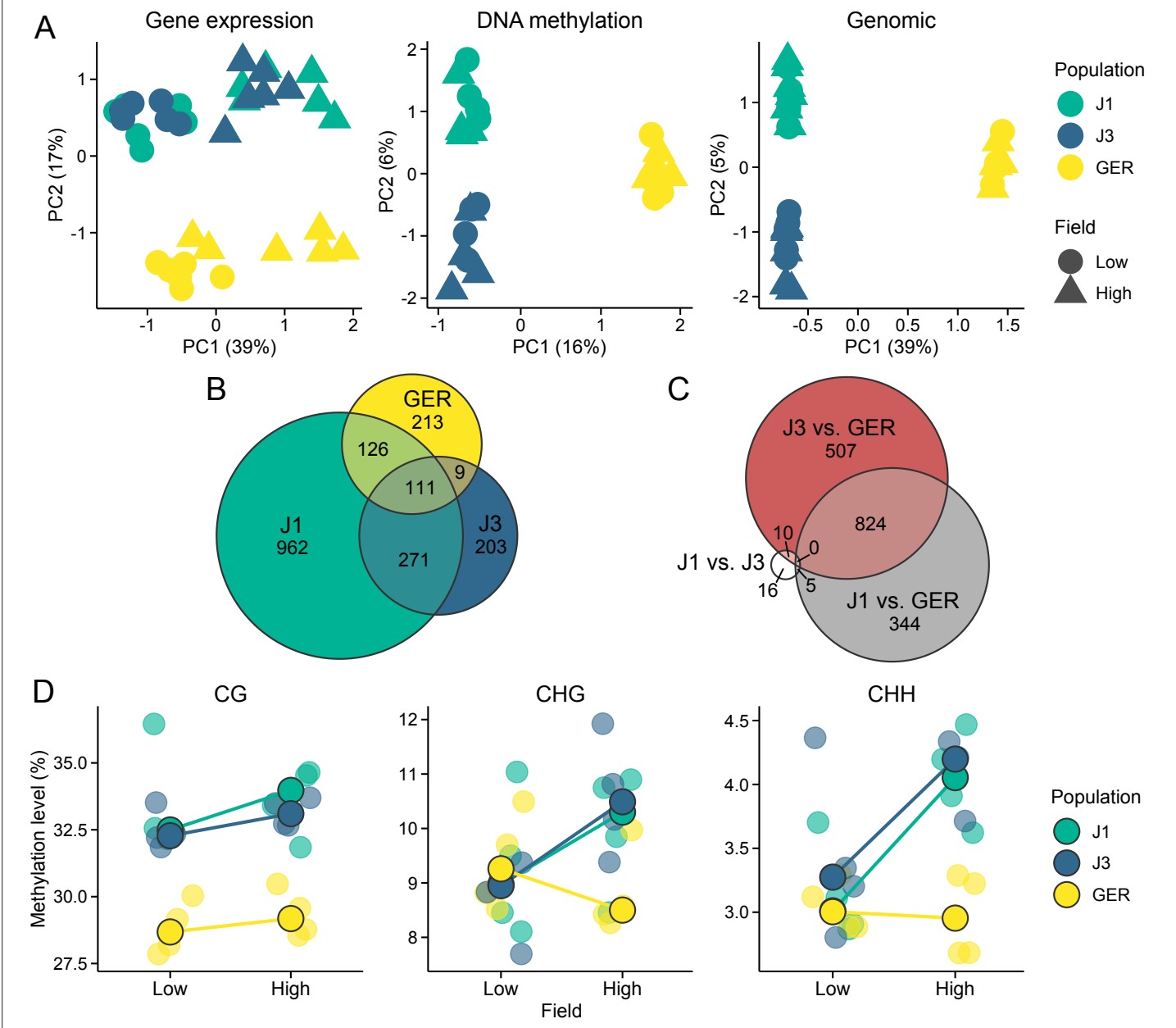

**Figure 2.** Gene expression and methylation variation across the field sites and populations. (**A**) Gene expression, DNA methylation, and genomic (based on SNPs called from the transcriptome data) variation along the first two eigenvectors of a principal component analysis (PCA). The proportion of variance explained by the principal components (PCs) is shown in parentheses. (**B**) The number of differentially expressed genes (DEGs) between field sites. (**C**) The number of DEGs between populations (across both fields). (**D**) Average methylation levels at the two field sites, shown for CG, CHG, and CHH contexts. Black-lined circles show median estimates for the populations, while individual estimates are shown with transparent colors in the background. Note the difference in y-axis scales between the panels.

The online version of this article includes the following figure supplement(s) for figure 2:

**Figure supplement 1.** Methylation variation along the first two eigenvectors of a principal component analysis (PCA), shown for the three methylation contexts.

**Figure supplement 2.** Overlap between candidate gene groups.

**Figure supplement 3.** The $\log_2$ OR of association between differentially expressed genes (DEGs) and differentially methylated regions 1 kb upstream of each gene, shown for each methylation context.

**Figure supplement 4.** Top five enriched (Q<0.05, hypergeometric test) gene ontology (GO) terms among each candidate gene set.

**Table 1.** Models for likelihood-ratio tests (LRTs) and the number of identified differentially expressed genes (DEGs) and differentially methylated genes (DMGs).

| Gene set | Full and reduced model | Number |
|---|---|---|
| DEG ~ field | Expression ~ population + field<br>Expression ~ population | 3456 |
| DEG ~ population | Expression ~ field + population<br>Expression ~ field | 1476 |
| DEG ~ field:population | Expression ~ field + population + field:population<br>Expression ~ field + population | 477 |
| DMG-CG ~ field | CG methylation ~ CH% + population + field<br>CG methylation ~ CH% + population | 112 |
| DMG-CG ~ population | CG methylation ~ CH% + field + population<br>CG methylation ~ CH% + field | 641 |
| DMG-CG ~ field:population | CG methylation ~ CH% + field + population + field:population<br>CG methylation ~ CH% + field + population | 260 |
| DMG-CH ~ field | CH methylation ~ population + field<br>CH methylation ~ population | 1036 |
| DMG-CH ~ population | CH methylation ~ field + population<br>CH methylation ~ field | 3580 |
| DMG-CH ~ field:population | CH methylation ~ field + population + field:population<br>CH methylation ~ field + population | 680 |

CH = CHG and CHH.
CH% = methylation rate at CHG and CHH contexts.

–0.21; $p < 2 \times 10^{-16}$), DEG ~ field shared fewer than expected genes with the DMG groups (*Figure 2—figure supplement 2*). Besides methylation of gene bodies, changes in gene expression may result from methylation of the promoter regions (*Law and Jacobsen, 2010*). To explore this, we compiled methylation data from 1 kb upstream of each gene and searched for an overlap between differentially methylated promoter regions and DEGs. For DEG ~ field and DEG ~ field:population, these results did not deviate from random expectations. DEG ~ population, by contrast, was associated with differential methylation at both gene bodies and promoter regions (*Figure 2—figure supplement 3*).

Based on gene ontology (GO) terms, most DEG and DMG sets were enriched for genes involved in specific biological processes (*Figure 2—figure supplement 4*). DEG ~ field, a group of genes with plastic expression responses, had multiple enriched GO terms involved in photosynthesis ('photosynthesis,' 'chlorophyll biosynthetic process,' and 'reductive pentose-phosphate cycle' as the top three terms). Among the top five terms were also 'response to cold' and 'response to light intensity,' which were previously found as enriched terms among local adaptation candidates in J3 (*Hämälä and Savolainen, 2019*). Genes showing genetic expression responses (DEG ~ population) were enriched for only three GO terms, two of which were related to defense against pathogens ('defense response to bacterium and virus'). DEG ~ field:population showed enrichment for multiple different processes, including 'response to cold.' Although we found fewer enriched GO terms among the DMGs, terms involved in cellular signaling and transport (e.g. 'signal transduction' and 'cation transport') were strongly represented among the non-CG DMGs (*Figure 2—figure supplement 4*).

## Footprints of selection at candidate gene sets

To examine selective signals at DEGs and DMGs, we used whole-genome sequence data from independent J1, J3, and GER individuals (*Hämälä et al., 2018*; *Mattila et al., 2017*; *Takou et al., 2021*). As environmental effects in CG methylation were rare, we primarily characterize selection at DEGs and non-CG DMGs. Compared to the genome-wide average, genes belonging to the DEG groups harbored lower than expected nucleotide diversity within populations as well as higher than expected nucleotide differentiation between populations (*Figure 3* and *Figure 3—figure supplement 1*), indicative of purifying selection or selective sweeps. In particular, the promoter regions at DEG ~

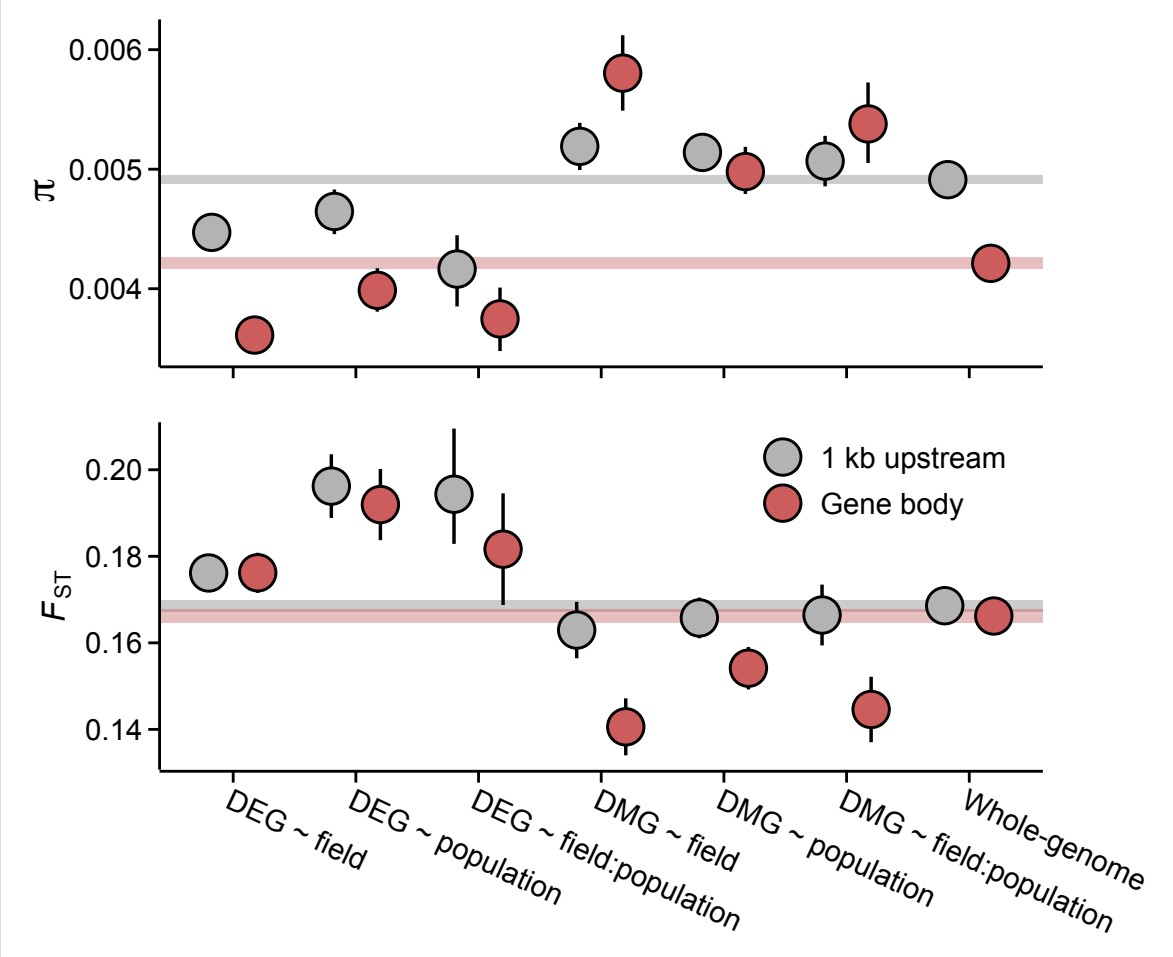

**Figure 3.** Pairwise nucleotide diversity (π) and $F_{ST}$ at the candidate gene sets. π estimates are shown for J3 (n=22) population. Estimates for J1 (n=9) and GER (n=17) are shown in *Figure 3—figure supplements 1 and 3*. $F_{ST}$ was estimated across the three populations. See *Figure 3—figure supplement 3* for pairwise $F_{ST}$ estimates. Error bars show 95% bootstrap-based CIs. Shaded areas mark the 95% CIs across all genes.

The online version of this article includes the following figure supplement(s) for figure 3:

**Figure supplement 1.** Pairwise nucleotide diversity (π) at the candidate gene sets.

**Figure supplement 2.** Population effects at DEG ~ field:population genes.

**Figure supplement 3.** Pairwise $F_{ST}$ estimates for the candidate gene sets.

**Figure supplement 4.** Pairwise nucleotide diversity (π) at CG and non-CG differentially methylated genes (DMGs).

field:population genes exhibited a combination of low genetic diversity and high differentiation, suggesting that parts of the transcriptional machinery, such as cis-regulatory elements, may have been subject to differential selection. The signals of expression and sequence differentiation were, however, almost exclusively found between the Norwegian populations and GER (*Figure 3—figure supplements 2 and 3*). In contrast to DEGs, non-CG DMGs harbored higher than average levels of genetic diversity (diversity levels at CG DMGs were lower and deviated less from the whole-genome average; *Figure 3—figure supplement 4*; *Figure 3*), potentially reflecting relaxed selective constraint. In the rest of the paper, the abbreviation DMG refers to non-CG DMGs.

The estimates of genetic diversity indicate that selective processes have shaped the nucleotide composition of the candidate gene sets. To study this in more detail, we inferred the strength of selective constraint on the coding sequence of DEGs and DMGs by modelling the distribution of fitness effects (DFE) of new nonsynonymous mutations (*Kim et al., 2017*) and by estimating the rate of adaptive evolution (α) (*Messer and Petrov, 2013*). In all three populations, genes belonging to the DEG ~ field and DEG ~ field:population groups appeared under strong purifying selection: fewer

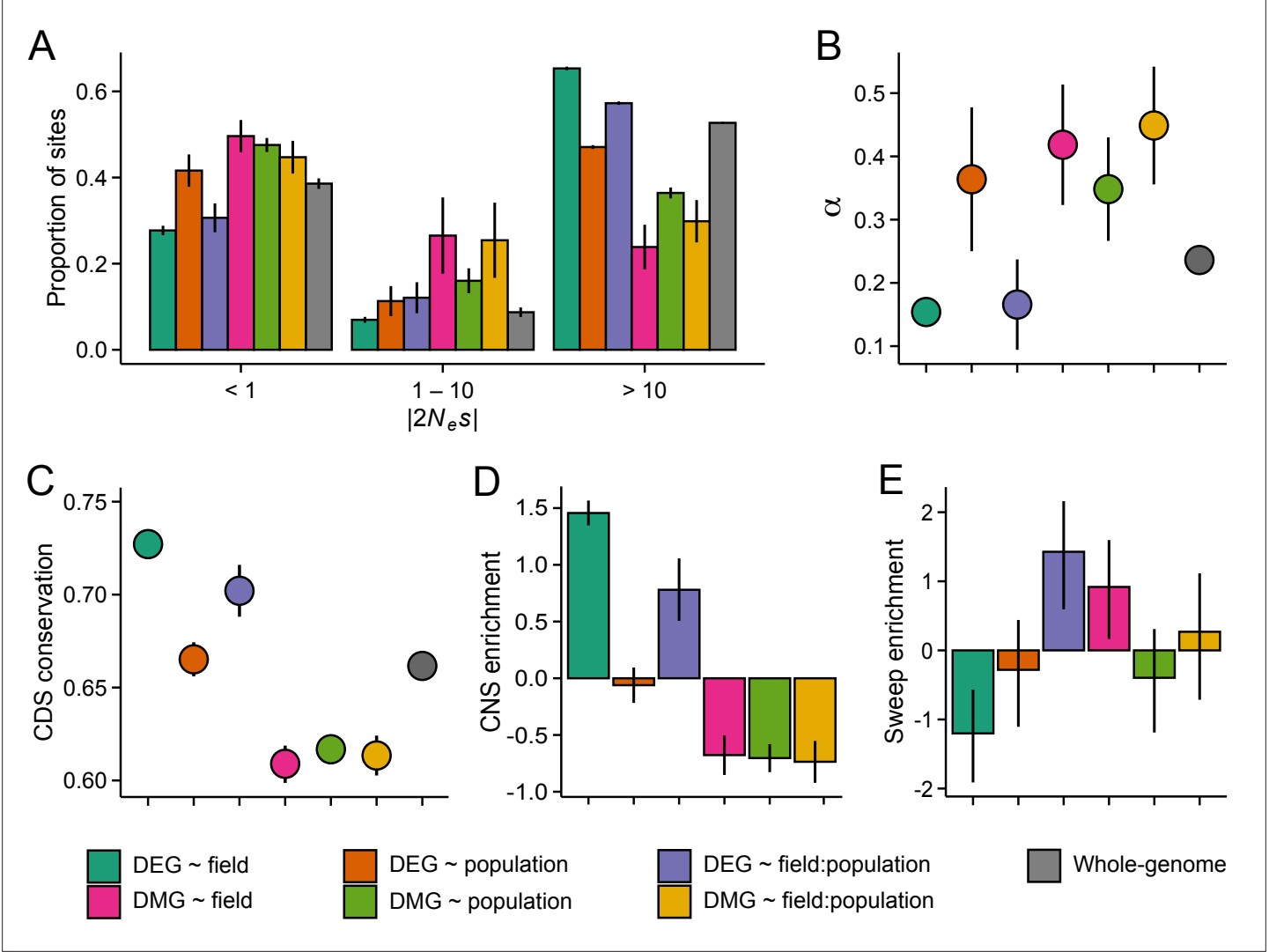

**Figure 4.** The efficacy of negative and positive selection at the candidate gene sets. (**A**) The distribution of fitness effects of new nonsynonymous variants. The mutations were divided into three bins based on the strength of purifying selection ($2N_es$): nearly neutral, intermediate, and deleterious, respectively. (**B**) The proportion of sites fixed by positive selection ($\alpha$). (**C**) The conservation of coding sequence (CDS). Shown are average GERP scores estimated for each gene group. The scores were rescaled from 0 to 1 using the range of possible values at each site. (**D**) The enrichment of conserved noncoding sequence (CNSs) 1 kb upstream of candidate genes. Shown are the $\log_2$ OR of association between CNSs and the candidate gene sets. (**E**) The enrichment of selective sweeps at the candidate genes. Shown are the $\log_2$ ORs of association between selective sweeps and the candidate gene sets. Panels A, B, and E show results for the J3 population (n=22). See *Figure 4—figure supplements 1 and 2* for results on J1 (n=9) and GER (n=17), respectively. For all panels, error bars show 95% bootstrap-based CIs.

The online version of this article includes the following figure supplement(s) for figure 4:

**Figure supplement 1.** The efficacy of negative and positive selection at candidate gene sets in the J1 population (n=9).

**Figure supplement 2.** The efficacy of negative and positive selection at candidate gene sets in the GER population (n=17).

than expected mutations were predicted to be nearly neutral ($2N_es$ <1) and more predicted to be deleterious ($2N_es$ >10) (*Figure 4A* and *Figure 4—figure supplements 1A and 2A*; p=0.004). If the strong selective constraint is due to localized increase of $N_e$, we might expect a similar increase in the efficacy of positive selection. However, $\alpha$ estimates indicated that fewer than expected nonsynonymous mutations have been fixed by positive selection in the DEG ~ field and DEG ~ field:population groups (*Figure 4B*, *Figure 4—figure supplements 1B and 2B*; p<0.004, LRT). In contrast to the plastic DEG groups, the DFE and $\alpha$ estimates indicated relaxed evolutionary constraint at other candidate gene groups. In particular, the DMGs had higher than expected $\alpha$ in all three populations

(*Figure 4B*, *Figure 4—figure supplements 1B and 2B*; p<0.0002, LRT), suggesting long-term adaptive evolution. Together, these results are indicative of reduced evolutionary rates at the plastic DEG groups as well as rapid molecular evolution at the DMGs. Both results were further supported by an analysis of coding sequence conservation among 26 eudicot species, which identified higher than expected conservation at the two DEG groups and lower than expected conservation at the DMGs (*Figure 4C*; p<0.0005, Wilcoxon rank-sum test). Furthermore, by searching for conserved noncoding sequences (CNSs) 1 kb upstream of each gene (*Haudry et al., 2013*), we found that CNSs were overrepresented at the DEG ~ field and DEG ~ field:population genes and underrepresented at the DMGs (*Figure 4D*), indicating that the promotor regions are similarly (un)conserved.

To explore how the differing evolutionary rates influence the short-term adaptive potential of these gene groups, we used RAiSD (*Alachiotis and Pavlidis, 2018*) to search for footprints of recent positive selection at each gene (gene body ±2 kb). We found that DEG ~ field exhibited a clear deficit of selective sweeps (*Figure 4E* and *Figure 4—figure supplements 1C and 2C*; p<0.006, Fisher's exact test), consistent with the reduced signals of adaptive evolution. By contrast, DEG ~ field:population showed an enrichment of selective sweeps in all three populations (*Figure 4E* and *Figure 4—figure supplements 1C and 2C*; p<0.002, Fisher's exact test). Between 10 and 16 genes overlapped the sweep regions in the three populations, with 8 genes shared between the populations. At each of these genes, windows producing the strongest selection signals were found upstream of the transcription start site (TSS), suggesting that the promotor regions may have been subject to recent positive selection.

## Environmental response at TEs

Besides genes, stressful environments may invoke plastic responses at TEs (*Capy et al., 2000*). To examine this in our data, we quantified TE methylation and TE expression. In contrast to genes, we found that methylation levels across TEs were clearly influenced by the field sites (*Figure 5A*). Compared to the low-altitude field, TEs at the high-altitude field site had higher methylation levels in all three cytosine contexts, but CHH methylation responded most strongly to the environment (*Figure 5A* and *Figure 5—figure supplements 1–3*; logistic regression $\beta_{CG}$ = 0.07, $\beta_{CHG}$ = 0.12, and $\beta_{CHH}$ = 0.17; p<2 × 10⁻¹⁶, LRT). Population effects were also evident in TE methylation: J1 and J3 exhibited clearer environmental responses compared to GER (*Figure 5A* and *Figure 5—figure supplements 1–3*). As methylation is commonly involved in epigenetic silencing of TEs (*Lisch, 2009*), we explored a connection between TE methylation and TE expression. Consistent with the notion of TE silencing, expression levels were negatively correlated with methylation levels (Spearman's $\rho_{CG}$ = –0.28, $\rho_{CHG}$ = –0.16, and $\rho_{CHH}$ = –0.14; p<6 × 10⁻¹⁰). However, increased methylation at the high-altitude field site did not lead to a large-scale silencing of TEs, as expression levels of both retrotransposons and DNA transposons were lower in the low-altitude field site (*Figure 5B* and *Figure 5—figure supplement 4*; p<2 × 10⁻¹⁶, Wilcoxon rank-sum test). By comparing the TE results to randomly compiled gene sets of equal size, we found that the observed expression differences are unlikely caused by technical bias (*Figure 5—figure supplement 5*). Therefore, rather than DNA methylation proactively suppressing TEs in the more stressful environment, these patterns suggest that hypermethylation at the high-altitude field site is a response to increased TE activation (*Lisch, 2009*; *Lloyd and Lister, 2022*). Furthermore, increased TE activation can directly influence gene expression, as genes whose promoters lie close to TEs may also get silenced by the DNA methylation (*Hollister and Gaut, 2009*; *Horvath and Slotte, 2017*; *Wyler et al., 2020*). Indeed, by examining the association between gene's expression level and its distance from a methylated TE, we found evidence that epigenetic regulation directed at TEs suppresses the expression of nearby genes (*Figure 5—figure supplement 6*). TE methylation did not, however, appear to underlie the expression differences observed between the field sites (or vice versa, *Secco et al., 2015*), as we found that DEG ~ field genes were, on average, further away from differentially methylated TEs than other genes (683 bp vs. 214 bp; p = 3.9 × 10⁻⁸, Wilcoxon rank-sum test).

Taken together, the methylation and expression results are consistent with broad-scale activation of TEs at the high-altitude field site. To explore whether such activation can shape genome evolution, we identified TE insertion polymorphisms from the whole-genome data. Based on allele frequency spectra summarized with Tajima's D (*Tajima, 1989*), TEs segregate at lower frequencies in J3 than in J1 (*Figure 5C*). Given that TEs generally have negative fitness consequences (*Bourgeois and Boissinot,*

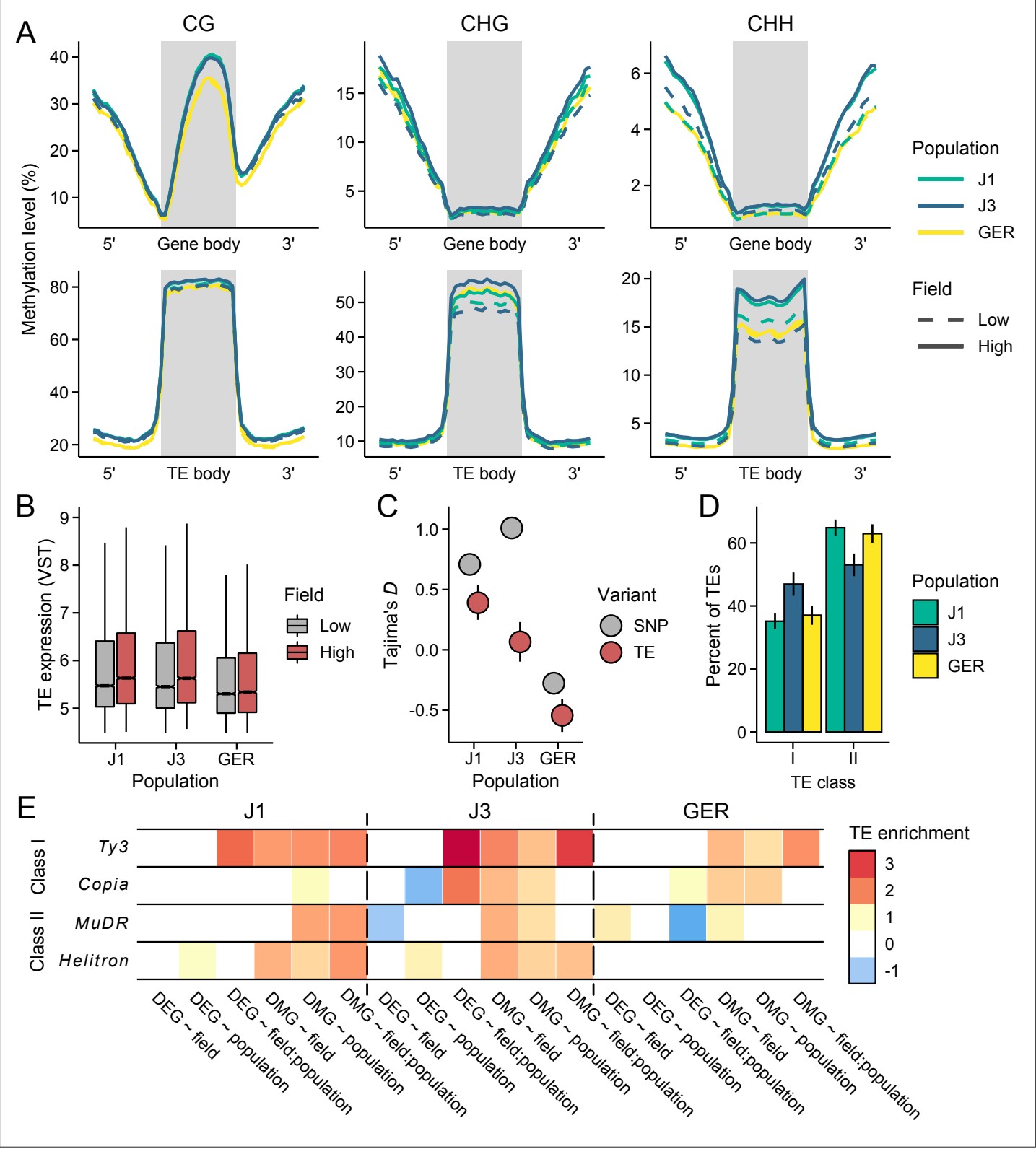

**Figure 5.** Environmental response at transposable elements (TEs). (**A**) Methylation levels across meta-genes and meta-TEs, shown for CG, CHG, and CHH contexts. Note the difference in y-axis scales between the panels. See *Figure 5—figure supplements 1–3* for results on different TE superfamilies (**B**) The expression of TE families at low- and high-altitude field sites. See *Figure 5—figure supplement 4* for results on different TE superfamilies. (**C**) Tajima's *D* for TEs and nonsynonymous SNPs in the three populations. (**D**) The proportion of retrotransposons (class I) and DNA transposons (class

*Figure 5 continued on next page*

*Figure 5 continued*

II) in each population. See *Figure 5—figure supplement 8* for results on different superfamilies. (**E**) Enrichment of TEs at gene bodies of differentially expressed genes (DEGs) and non-CG differentially methylated genes (DMGs). Shown are the $\log_2$ ORs of association between the four largest TE superfamilies and the candidate gene sets ($Q<0.05$). See *Figure 5—figure supplement 10* for results on all TEs, including up- and downstream regions of genes. For C and D panels, error bars show 95% bootstrap-based CIs.

The online version of this article includes the following figure supplement(s) for figure 5:

**Figure supplement 1.** CG methylation levels of different transposable element (TE) superfamilies at low- and high-altitude field sites.

**Figure supplement 2.** CHG methylation levels of different transposable element (TE) superfamilies at low- and high-altitude field sites.

**Figure supplement 3.** CHH methylation levels of different transposable element (TE) superfamilies at low- and high-altitude field sites.

**Figure supplement 4.** Expression levels of different transposable element (TE) superfamilies at low- and high-altitude field sites.

**Figure supplement 5.** Observed transposable element (TE) expression difference between the field sites compared to 10,000 randomly compiled gene sets of equal size.

**Figure supplement 6.** The association between gene expression and distance from the closest transposal element (TE; upstream of transcription start site [TSS]).

**Figure supplement 7.** Efficacy of selection in the three populations.

**Figure supplement 8.** The proportion of different transposal element (TE) superfamilies in each population.

**Figure supplement 9.** The count of different transposal element (TE) superfamilies in each population.

**Figure supplement 10.** The $\log_2$ OR of association between transposal elements (TEs) and the candidate gene sets.

*2019*), the lower frequency could be due to more efficient purging of TEs in J3. However, based on SNP data, the efficacy of selection in J3 appears equal (or weaker) to J1 (*Figure 5C* and *Figure 5— figure supplement 7*), pointing toward an alternative explanation: a recent influx of TE insertions can result in an excess of rare variants, as not enough time has passed for neutral or slightly deleterious TEs to increase in frequency (*Bergman and Bensasson, 2007*). Furthermore, if TE activation has resulted in an excess of recent TE insertions, we might expect different effects on retrotransposons and DNA transposons; as retrotransposons move by 'copy-and-paste' mechanism and DNA transposons move by 'cut-and-paste' mechanism (*Wicker et al., 2007*), the number of new retroelements likely increases more rapidly as a result of such activation. As the probability of TEs impairing gene function is increased along with their activation, the number of active DNA transposons could also be more effectively reduced by purifying selection. Consistent with these expectations, we found an enrichment of LTR retrotransposons and a deficit of DNA transposons in J3 compared to J1 (*Figure 5D* and *Figure 5—figure supplements 8 and 9*). Therefore, our results suggest that stress-induced TE activation could have resulted in recent proliferation of retrotransposons (and potentially purging of DNA transposons) in the alpine population, although we cannot rule out the role of drift in shaping the composition of the TE landscape. Finally, by examining the locations of the TE insertions, we discovered that many genes belonging to the six candidate groups were more likely to harbor TE insertions than the average gene (*Figure 5E* and *Figure 5—figure supplement 10*). In particular, the DEG ~ field:population group was highly enriched for *Ty3* retrotransposons in the J3 population ($p=1.3 \times 10^{-6}$, Fisher's exact test). We also found that DMGs were TE insertion hotspots (*Figure 5E*), suggesting that many of the non-CG methylation changes may have arisen as a result of TE methylation spreading to genic regions.

## Discussion

### Gene expression is strongly influenced by the environment

We conducted a reciprocal transplant experiment to study environmentally induced variation in gene expression and DNA methylation in three populations of *A. lyrata*. Our study had the benefit of exposing the plants to full extent of environmental stressors for an entire year, likely proving a more realistic view of transcriptome and DNA methylome responses in the face of environmental change than laboratory experiments (*Savolainen et al., 2013*). We found that gene expression variation was strongly plastic, as we detected many more DEGs between the field sites than between populations. In particular, the two Norwegian populations had highly similar transcriptional responses, likely owing to their close geographical and genetic similarity (diverged ~1700 years ago, *Hämälä*

*et al., 2018*). However, we previously found evidence of local adaptation between these populations (*Hämälä et al., 2018*), demonstrating that some ecologically important traits are genetically differentiated despite the strong plasticity (e.g. flowering time, *Figure 1C*). In fact, if such differentiation is primarily manifested at earlier life history stages, our sampling after 1 year likely missed those effects. Compared to the Norwegian populations, GER exhibited distinct expression responses, consistent with its greater genetic dissimilarity (diverged from the Norwegians ~150,000 years ago, *Takou et al., 2021*) and different growing environment (*Leinonen et al., 2009*). Given that GER plants fared poorly at both field sites during our multi-year experiment (*Hämälä et al., 2018*), we may assume that much of the expression differentiation in GER is maladaptive in the Norwegian environment. Nevertheless, as such differentiation can arise from evolved effects in the local populations (i.e. for some genes GER exhibits the ancestral expression), comparing the Norwegian populations to GER likely revealed adaptive expression responses. Such differences may have further been amplified by the timing of our sampling (late August), as the local and nonlocal plants likely differed in their physiological preparation for winter.

## The DNA methylome is largely insensitive to the environment

In contrast to gene expression, variation in DNA methylation was mainly shaped by population history, and environmental effects were largely restricted to TEs. As such, our experiment revealed little evidence of environmentally induced DNA methylation influencing gene expression. Studies conducted in controlled conditions have discovered similar results in the sister species *A. thaliana*; patterns of DNA methylation have been weakly influenced by temperature (*Dubin et al., 2015*), phosphate starvation (*Secco et al., 2015*), salt stress (*Wibowo et al., 2016*), drought stress (*Ganguly et al., 2017*), and light intensity (*Ganguly et al., 2018*). Taken together, these results suggest that the DNA methylome in *Arabidopsis* species is largely insensitive to environmental factors and thus unlikely to induce large-scale changes in gene expression within a single generation. We note, however, that here, we only studied DNA methylation, whereas other epigenetic modifications might be more readily induced by the environment. For example, as histone modifications are known to exert a major influence on gene expression (*Lloyd and Lister, 2022*), future work studying environmental induction of histone modifications in natural conditions holds promise to expand our understanding of the evolutionary consequences of epigenome plasticity (*Husby, 2022*). In any case, we discovered greater than expected overlap between genes that were differentially expressed and methylated between the populations, suggesting that changes in DNA methylation, when they accumulate over generations, may influence patterns of gene expression (*Muyle et al., 2022*; *Takuno et al., 2017*).

## Environmentally responsive genes evolve under strong selective constraint

We found that environmentally induced expression responses were highly consistent across our experimental plants, indicating that the underlying regulatory elements are largely conserved between the three *A. lyrata* populations. Previous studies examining environmentally responsive genes have found that the regulatory conservation is frequently associated with the conservation of the gene product (*He et al., 2021*; *Hodgins et al., 2016*; *Hunt et al., 2013*; *Lasky et al., 2014*; *Lowry et al., 2013*; *Zhang et al., 2017*). Although our results qualitatively support these findings, as genes showing consistent expression responses to the environment (DEG ~ field) were under stronger purifying selection than those showing genetically variable responses (DEG ~ field:population), we discovered that both gene sets exhibited stronger than expected signals of evolutionary constraint. By examining nucleotide diversity and differentiation upstream of the TSS, we further found that population-specific selection at promoter regions (potentially at cis-regulatory variants) has likely contributed to the maintenance of heritable expression variation at the DEG ~ field:population genes. We note, however, that our experimental design did not allow us to associate selective signals with trans-acting variants, which may have also played a role in the observed expression differentiation (*Hämälä et al., 2020*; *Lopez-Arboleda et al., 2021*). Overall, our results suggest that genes whose products are involved in conserved processes are prone to exhibit plastic expression responses to environmental stressors, but genetic variation in the environmental response still arises through changes at regulatory elements. However, the expression differentiation was almost exclusively found between the Norwegian

populations and GER, indicating that such rewiring of the regulatory network may primarily happen over long evolutionary timescales.

## Plastic responses at TEs create novel genetic variation

Although DNA methylation at genic regions was largely insensitive to the environment, we discovered clear environmental effects in TE methylation. TEs at the high-altitude field site had increased methylation and expression levels, suggestive of stress-induced TE activation. Compared to the lowland population, plants native to the alpine environment harbored an excess of rare TE insertions, suggesting that certain TE families, particularly LTR retrotransposons, may have recently expanded their copy numbers. On average, such TE mobilization is likely to have detrimental effects, as TEs can impair gene function by either directly disrupting coding regions and regulatory elements (*Bourgeois and Boissinot, 2019*) or by influencing the expression of nearby genes through the spread of epigenetic silencing (*Hollister and Gaut, 2009*; *Horvath and Slotte, 2017*; *Wyler et al., 2020*). Indeed, we also discovered that genes whose promoters lie close to the methylated TEs had reduced expression levels. On the other hand, TE mobility is expected to create novel genetic variants that may, in some cases, facilitate adaptation under new selective environments (*Capy et al., 2000*; *Casacuberta and González, 2013*; *McClintock, 1984*).

Here, we found that genes exhibiting genetically variable responses to the environment harbored an enrichment of LTR retrotransposons in the alpine population. Although we lack functional validation for the detected TEs, such enrichment demonstrates that TEs have the potential to create new heritable variation that is relevant for environmental adaptation. The role of TEs in creating novel functional variants is also seen in results by *Quadrana et al., 2019*, who found experimental evidence that retrotransposons of the *Copia* superfamily preferentially integrate into the bodies of environmentally responsive genes in *A. thaliana*. This integration is likely mediated by the histone variant H2A.Z, which is commonly found within the gene bodies of temperature sensing genes (*Talbert and Henikoff, 2014*), providing a possible mechanism for the observed TE accumulation in the J3 population. Indeed, by searching the *A. lyrata* genome for H2A.Z binding sites defined in *A. thaliana* (*Chow et al., 2019*), we confirmed that the DEG ~ field:population genes were more likely to carry such sites than the average gene (OR = 3.10; p<2 × 10$^{-16}$, Fisher's exact test). Interestingly, the DEG ~ field genes were also enriched for H2A.Z binding sites (OR = 2.93; p<2 × 10$^{-16}$, Fisher's exact test), suggesting that these genes may be similarly targeted by TE insertions, but due to stronger selective constraint, TEs are purged more efficiently. However, although we also detected an enrichment of *Copia* elements in the environmentally responsive genes, the enrichment of *Ty3* elements was far greater (OR: 4.5 vs. 11.7). This pattern indicates that the *Ty3* superfamily exhibits a similar integration preference in *A. lyrata* and/or that *Copia* elements have been more readily purged by purifying selection. If similar dynamics exist in other Brassicaceae species, it could explain the apparent excess of *Ty3* TEs in the alpine species *Arabis alpina* and *Draba nivalis* (*Nowak et al., 2021*).

## Ideas and speculation

By studying gene expression and DNA methylation in natural conditions, we have gained new insights into evolutionary forces shaping plastic and genetic variation. We observed contrasting results between the molecular phenotypes, as gene expression was primarily influenced by the environment, and DNA methylation was primarily influenced by population history. We found that the coding and promoter regions of most environmentally responsive genes are slowly evolving, which could influence the adaptive potential of these *A. lyrata* populations by keeping the pool of segregating variants small. In particular, as adaptation to rapidly shifting fitness optima can be limited by the lack of standing genetic variation (*Matuszewski et al., 2015*), the strong selective constraint and low genetic diversity at environmentally responsive genes could pose a risk in cases where environmental perturbations exceed the buffering mechanism provided by phenotypic plasticity (*Ghalambor et al., 2007*; *Price et al., 2003*). However, one caveat with this interpretation is that mutations in large-effect trans-loci could lead to wide-ranging expression effects despite the conservation of the cis-regulatory elements (*Josephs et al., 2020*; *Lopez-Arboleda et al., 2021*). Furthermore, if the genetic architecture of adaptation is much more polygenic than detected here, the selective constraint would likely offer less resistance to evolutionary responses (*Hayward and Sella, 2022*; *Stetter et al., 2018*). Although epigenetic modifications have been proposed as a mechanism of coping with rapid environmental

change (*McGuigan et al., 2021*), we found little evidence that environmentally induced changes in the DNA methylome could influence adaptive phenotypes in *A. lyrata*. Our results further suggest that novel heritable variation may be rapidly created by TEs integrating into the gene bodies of environmentally responsive genes, but further functional validation would be required to determine the phenotypic and fitness effects of such variants.

## Materials and methods

### Reciprocal transplant experiment and sample collection

We conducted a reciprocal transplant experiment to study altitude adaptation among Norwegian populations of *A. lyrata* ssp. *petraea*. *A. lyrata* is a predominantly outcrossing, perennial herb with a wide circumpolar distribution across the northern hemisphere (*Jalas and Suominen, 1994*). It is closely related to the model species *A. thaliana*, but differences in key life history traits (*A. thaliana* is a highly selfing annual) make *A. lyrata* a useful study system in ecology and evolution (*Savolainen and Kuittinen, 2011*). Detailed experimental designs are presented in *Hämälä et al., 2018*, and here, we give a brief explanation of the relevant methodology. Seed material was collected from two locations around the Jotunheimen national park, Lom (61.84°N, 8.57°E; altitude 300 m.a.s.l.) and Spiterstulen (61.62°N, 8.40°E; altitude 1100 m.a.s.l.). We additionally used an *A. lyrata* population from Germany (49.65°N, 11.48°E; altitude 400 m.a.s.l.) as a comparison group. In *Hämälä et al., 2018*, the Jotunheimen alpine region was represented by four populations, which were abbreviated as J1–J4. To keep the naming convention consistent with previous work (*Hämälä et al., 2018*; *Hämälä and Savolainen, 2019*), we refer to these populations as J1 (Lom), J3 (Spiterstulen), and GER (Germany).

Plants were initially grown in a growth chamber and crossed to produce full-sib families for each population. The crossing progeny was germinated and pre-grown in a greenhouse at University of Oulu, Finland for about 2 months. In August 2014, we established two experimental fields in Jotunheimen, Norway: a low-altitude field site in Lom (300 m.a.s.l.), close to the natural growing environment of J1, and a high-altitude field site in Spiterstulen (1100 m.a.s.l.), close to the natural growing environment of J3 (*Figure 1A*). Using multi-year fitness measurements, we found evidence of home-site advantage between the J1 and J3 populations, whereas GER had poor fitness at both field sites (*Hämälä et al., 2018*). After 1 year, we chose six individuals per population from both fields (a total of 36 individuals) for sample collection and sequencing. Individuals from both Norwegian populations represented six seed families, with individuals from the same families collected from both fields, whereas GER was represented by 10 seed families (due to high mortality, many GER seed families were not available in both fields). Leaf samples were collected during two consecutive days (August 21 and 22, 2015, between 18:30 and 19:30), immersed in RNAlater stabilizing solution (Thermo Fisher Scientific), and kept at –20°C until library preparation.

### Transcriptome sequencing

Total RNA was extracted using a RNeasy Plant Mini kit (QIAGEN), following the manufacturer's instructions. RNA integrity number and concentration were determined using an Agilent RNA 6000 Pico kit (Agilent). NEBNext Poly(A) mRNA Magnetic Isolation Module (New England Biolabs, NEB#E7490) from NEBNext Ultra Directional RNA Library Prep kit for Illumina was used to create stranded RNA-seq libraries according to the manufacturer's instructions. RNA-seq libraries were quantified using KAPA Library Quantification kits (Kapa Biosystems) in combination with an Agilent High Sensitivity DNA kit (Agilent Genomic). The RNA-seq libraries were sequenced on four lanes of Illumina HiSeq2000 (paired-end 100 bp) at the Institute for Molecular Medicine Finland (FIMM), University of Helsinki.

The transcriptome sequencing yielded a total of 1.036 billion read pairs, with an average of 29 million read pairs per individual (*Supplementary file 1*). After removing low quality reads and sequencing adapters with Trimmomatic (*Bolger et al., 2014*), we used STAR (*Dobin et al., 2013*) to align reads to the *A. lyrata* reference genome (*Hu et al., 2011*) and to count reads mapping uniquely to each gene model. To quantify TE expression, we first identified TEs using RepeatMasker (https://www.repeatmasker.org/) and a library of *A. lyrata* consensus TEs from RepetDB (*Amselem et al., 2019*), covering the main orders of both retrotransposons and DNA transposons (we ignored pseudogenes and TEs without classification). This resulted in the discovery of 80,624 individual TEs across the eight main chromosomes. Following *Anderson et al., 2019*, we characterized the expression of different TE

families using both uniquely and nonuniquely mapped reads. To do so, we used mmquant (*Zytnicki, 2017*) to count reads mapping to individual TEs and compiled count data for each TE family. We only included reads that mapped uniquely to a single element or to multiple elements belonging to the same TE family (we allowed up to 665 multi-mapping locations in STAR, which was the maximum number of TEs from a single family). Reads that mapped to multiple TE families, as well as reads mapping to genes, were excluded. The TE expression data were then normalized by applying variance stabilization transformation (*Love et al., 2014*) to combined TE and gene counts.

## Whole-genome bisulfite sequencing

To examine patterns of DNA methylation, we used 24 samples (four individuals per population from both fields) for WGBS. DNA was extracted using a DNeasy Plant Mini kit (QIAGEN), following the manufacturer's instructions. We prepared the WGBS libraries according to MethylC protocol (*Urich et al., 2015*) and quantified them using a combination of KAPA Library Quantification kits (Kapa Biosystems) and an Agilent High Sensitivity DNA kit (Agilent Genomic). The WGBS libraries were sequenced on six lanes of Illumina HiSeq2000 (paired-end 100 bp) at FIMM, University of Helsinki.

The WGBS yielded a total of 768 million read pairs, with an average of 32 million read pairs per individual (*Supplementary file 1*). Low-quality reads and sequencing adapters were removed with Trimmomatic (*Bolger et al., 2014*), and the surviving reads were aligned to the *A. lyrata* reference genome (*Hu et al., 2011*) using Bismark (*Krueger and Andrews, 2011*) and Bowtie2 (*Langmead and Salzberg, 2012*). We removed duplicated reads using deduplicate_bismark script from Bismark and estimated the number of methylated and unmethylated reads at each cytosine context (CG, CHG, CHH, where H is A, T, or C) using a Bismark script bismark_methylation_extractor. We then removed sites with known C to T and A to G (the reverse orientation of C to T) SNPs from the methylation calls, because such SNPs will be incorrectly called as unmethylated by Bismark. To do so, we first identified SNPs using the RNA-seq data, which contained the same individuals as used for the WGBS (see below for details on SNP calling). For regions not covered by the RNA-seq in at least 50% of individuals, we used whole-genome sequencing data from independent J1 (n=9), J3 (n=22), and GER (n=17) individuals. By examining the methylation patterns of chloroplast DNA, which is expected to be naturally unmethylated, we estimated an overall conversion efficacy of 99.1% for CG context, 99.4% for CHG context, and 99.7% for CHH context (*Supplementary file 2*).

## Differential expression analysis

After confirming that sequencing batches are unlikely to bias the detection of DEGs (see Appendix 1 for details), we searched for DEGs between the field sites and populations using DESeq2 (*Love et al., 2014*). To identify population specific DEGs, we fit a single dispersion parameter to each gene with median read counts >1 and used the DESeq2's contrast function and Wald test to detect differences between the field sites. LRTs were used to identify global DEGs due to field site or population; we compared the fit of a full model, containing field, population, and their interaction as predictors, to a reduced model with one of the predictors removed (*Table 1*). To focus on top-ranking DEGs, we took a relatively stringent approach to account for multiple testing and required that outliers had a Bonferroni corrected p value<0.05. However, as DEGs showing an interaction between field site and population (DEG ~ field:population) only had 28 genes with Bonferroni corrected p<0.05, we also used a more lenient approach and required that outliers from the interaction tests had a false discovery rate-based Q value (*Storey, 2002*) below 0.05, leading to 477 DEGs. By being more lenient with multiple comparison, we reach a sample size needed to examine selective signals at these genes, although at the same time we risk accepting more false positives due to confounding factors.

## Differential methylation analysis

We removed sites with ≤3 reads and defined methylation levels as the proportion of unconverted cytosines. To associate the methylation patterns more closely with gene expression levels and measures of selective constraint, we primarily focused on methylation at gene bodies. Genic methylation can be separated into two classes: CG methylation, which tends to be associated with increased gene expression levels (outside transcription start and stop sites), and non-CG methylation (i.e. CHG and CHH), which tends to be associated with decreased gene expression levels (*Muyle et al., 2022*). We therefore searched for DMGs using data from the two methylation classes separately. Focusing on

cytosines with <50% missing data, we calculated the methylated (2 × the sum of methylation proportions) and unmethylated (2 × the sum of 1 – methylation proportions) allelic dosage for each individual at a given gene and tested the effects of the field site and population using logistic regression and LRTs. To find DMGs associated with CG methylation, we included non-CG methylation proportions as a cofactor in the models (*Table 1*). The LRT-based p values were transformed to Q values (*Storey, 2002*) to account for multiple testing. To define DMGs, we required that the genes had ≥10 cytosines and Q<0.05. Given that CG methylation was strongly influenced by population structure (*Figure 2A*), we corrected p values from the field + population to field LRTs (i.e. CG DMGs due to population) using genomic inflation factor (*Devlin and Roeder, 1999*).

## GO enrichment analysis

We conducted a GO enrichment analysis to identify biological processes associated with DEGs and DMGs. To do so, we used the direct *A. thaliana* orthologs of 21,784 genes to define GO terms and tested for an enrichment of biological processes (molecular functions and cellular components were ignored) using hypergeometric tests and Q-values (*Storey, 2002*). We then used REVIGO (*Supek et al., 2011*) to combine redundant GO terms.

## Whole-genome sequence data

To study the influence of selection on patterns of gene expression and DNA methylation, we used previously published whole-genome sequence data from J1 (n=9), J3 (n=22), and GER (n=17) individuals (*Hämälä et al., 2018*; *Mattila et al., 2017*; *Takou et al., 2021*). We removed low-quality reads and sequencing adapters with Trimmomatic (*Bolger et al., 2014*) and aligned the reads to the *A. lyrata* reference genome (*Hu et al., 2011*) with BWA-MEM (*Li, 2013*). We removed duplicated reads with Picard tools (https://broadinstitute.github.io/picard/) and realigned indels with GATK (*McKenna et al., 2010*).

To incorporate genotype uncertainty directly into our analyses, we used ANGSD (*Korneliussen et al., 2014*) to estimate genotype likelihoods and probabilities at each mono- and biallelic site. We used the GATK likelihood model and required reads to map uniquely, have mapping quality ≥30, and base quality ≥20. For each population, we had ANGSD estimate posterior genotype probabilities using allele frequency as a prior. To account for genetic structure in multi-population data, we estimated genotype probabilities using PCAngsd (*Meisner and Albrechtsen, 2018*), which employs a model that incorporates the effects of population structure (in the form of PCs) in the prior. We then estimated the allelic dosage, or the expected genotype, from the posterior probabilities as $\mathrm{E}\left[G\right] = \sum_{g=0}^{2} gP\left(G=g\right)$, where $G$ is the genotype. As the identification of selective sweeps (see below) required VCF files as input, we further used ANGSD to call genotypes from the posterior probabilities.

## Genetic diversity and differentiation

To assess how selection has acted on genes determined by their DEG and DMG status, we first estimated pairwise nucleotide diversity ($\pi$) and $F_{ST}$ for each candidate gene set. $\pi$ was estimated across all callable (variant and invariant) sites to avoid biasing the estimates by missing sites. We used the method by *Weir and Cockerham, 1984* to estimate $F_{ST}$ for both three- and two-population comparisons. We estimated $\pi$ and $F_{ST}$ for each candidate gene group, defined CIs for the estimates by resampling with replacement across genes 1000 times, and compared the estimates against $\pi$ and $F_{ST}$ calculated across all genes.

## Distribution of fitness effects

We examined the strength of selective constraint on the candidate gene sets by modelling the DFE. To do so, we estimated folded site frequency spectra (SFS) for synonymous (fourfold) and nonsynonymous (zerofold) sites using ANGSD (*Korneliussen et al., 2014*). We then used ∂a∂i (*Gutenkunst et al., 2009*) to fit three-epoch demographic models to synonymous SFS and inferred a deleterious gamma DFE for nonsynonymous sites, conditional on the demography, using fit∂a∂i (*Kim et al., 2017*). Population mutation rates ($\theta=4N_e\mu$) were estimated from the synonymous data and multiplied by 2.76 (*Takou et al., 2021*) to approximate $\theta$ at nonsynonymous sites. This approach, as opposed to optimizing $\theta$ along with the DFE parameters, can take into account variants expected to be missing

due to strong purifying selection (**Kim et al., 2017**). Demographic parameters were estimated from the whole-genome data and fixed for the analysis of candidate gene sets. We estimated CIs for the DFE parameters by fitting the models to 500 parametric bootstrap SFS. We assumed that counts in the bootstrap replicates followed a multinomial distribution with number of trials corresponding to total number of sites in the SFS and the probability of success corresponding to proportion of sites in a given allele frequency bin. We then discretized the DFE into three bins, nearly neutral ($2N_es <1$), intermediate ($1\le 2N_es \le10$), and deleterious ($2N_es >10$) and compared bins in the candidate gene sets to the genome-wide average. p Values were defined as twice the proportion of overlapping bootstrap replicates in each bin.

## The rate of adaptive evolution

We examined the efficacy of positive selection by estimating the proportion of sites fixed by positive selection ($\alpha$) (**Smith and Eyre-Walker, 2002**). Because $\alpha$ estimation requires outgroup information, we inferred unfolded SFS for synonymous and nonsynonymous sites using a method that accounts for uncertainty in the assignment of ancestral vs. derived alleles (**Keightley and Jackson, 2018**). We used *A. lyrata – A. thaliana – Capsella rubella – A. alpina* whole-genome alignments from **Hämälä and Tiffin, 2020**, requiring each site to have outgroup information in at least two of the three species. Following **Messer and Petrov, 2013**, we estimated $\alpha$ for each polymorphic allele frequency bin of the unfolded SFS as:

$$\alpha\left(x\right) = 1 - \frac{p_N\left(x\right)/p_S\left(x\right)}{d_N/d_S},$$

where $p_N\left(x\right)$ and $p_S\left(x\right)$ are the number of nonsynonymous and synonymous polymorphic sites at frequency $x$, and $d_N$ and $d_S$ are the number of nonsynonymous and synonymous fixes sites. We then used the R package nls2 (**Grothendieck, 2013**) to fit an asymptotic function of the form: $\alpha\left(x\right) = a + be^{-cx}$ to the data. As in **Haller and Messer, 2017**, we used the 'brute-force' algorithm to find suitable starting values for the free parameters (*a*, *b*, and *c*) and refined the values using a second step of standard nonlinear least-squares regression. We used LRTs to determine whether the fitted functions differ between the candidate gene sets and the genome-wide average and estimated CIs by fitting the models to 500 parametric bootstrap SFS.

## Sequence conservation

To examine the conservation of selective processes, we used GERP++ (**Davydov et al., 2010**) to estimate nucleotide conservation at DEGs and DMGs. We chose 25 eudicot species belonging to the clade Superrosidae (**Supplementary file 3**), whose divergence times in relation to *A. lyrata* ranged from 1.6 million years (*A. halleri*) to 123 million years (*Vitis vinifera*) (**Hohmann et al., 2015**). We first identified homologs by conducting BLAST queries against protein databases constructed for the species, keeping only the best match with alignment *e*-value $<1 \times 10^{-5}$ for each gene. For sets of homologs with ≥13 species, we aligned the coding sequences with MAFFT (**Nakamura et al., 2018**). As GERP++ requires an evolutionary tree, we used the R package phangorn (**Schliep, 2011**) to estimate a maximum likelihood tree across 1000 randomly selected genes with no missing species. Using the species tree and multiple alignments, we had GERP++ estimate the rejected substitutions (RS) score for sites in the *A. lyrata* coding sequence, quantifying the level of nucleotide conservation in relation to neutral substitution rate. Last, using the range of possible values at each site, we rescaled the RS scores from 0 to 1, where 0 is the weakest possible conservation and 1 is the strongest.

As our RS scores were estimated only for the coding sequence, we further used publicly available data on CNSs (based on the comparison of nine Brassicaceae species, **Haudry et al., 2013**) to assess nucleotide conservation at the promoter regions. To do so, we searched for the presence of CNSs 1 kb upstream of each gene and tested whether they are over- or underrepresented at our candidate gene groups using the Fisher's exact test.

## Scan for selective sweeps

To identify genes that have undergone recent selective sweeps, we used RAiSD (**Alachiotis and Pavlidis, 2018**) to scan the genomes for patterns of segregating sites, linkage-disequilibrium, and nucleotide diversity indicative of positive selection. We excluded regions that contained no

sequencing data, estimated the composite statistic µ in 50 SNP sliding windows, and characterized selective signals at each gene using the maximum µ value of windows within 2 kb. We then ran RAiSD on simulated neutral data (*Hämälä and Savolainen, 2019*) to find outliers. We considered observed estimates exceeding 99% of the simulated values as reflecting selective sweeps.

## Identification of TE insertion polymorphisms

We identified TE insertions polymorphisms using PoPoolationTE2 (*Kofler et al., 2016*). Following the recommended workflow, we masked the *A. lyrata* consensus TEs (*Amselem et al., 2019*) from the reference genome using an iterative mapping approach; simulated TE reads were aligned to the genome with BWA-MEM (*Li, 2013*), aligned regions masked from the reference with BEDtools (*Quinlan and Hall, 2010*), and the process repeated until no new unmasked regions were found. We then merged the consensus TEs with the masked genome, aligned quality-trimmed DNA-seq reads to the TE merged reference using BWA-MEM, and removed duplicated reads using Picard tools (https://broadinstitute.github.io/picard/). We only used samples with an average coverage ≥10 × across the eight main chromosomes (this excluded five individuals from J3 and two from GER) and required each site to have a minimum coverage of 6×. Furthermore, to compare how many TE insertions have accumulated in the three populations without biasing our estimates with different sample sizes and sequencing depths, we chose nine individuals from each population (i.e. the sample size of J1) and randomly sampled an equal number of aligned read pairs (20 million) from each individual. We then had PoPoolationTE2 estimate the proportion of reads supporting TEs in each individual and filtered the list to remove overlapping TEs. Last, we discretized the read proportions into genotypes: >0.85 TE homozygote, 0.85–0.15 TE heterozygote, and <0.15 non-TE homozygote.

## Acknowledgements

We thank M and L Bakkom for help with the field experiments, S Alatalo for help with sequencing library preparation, and M Takou, members of the Pyhäjärvi lab, the reviewing editor J Ross-Ibarra, and two anonymous reviewers for their comments on improving the manuscript. Computational resources were provided by CSC – Finnish IT Center for Science. This work was supported by the Academy of Finland's Research Council for Biosciences, Health and the Environment (decision 132611 to OS and 339702 to TH).

## Additional information

### Funding

| Funder | Grant reference number | Author |
| --- | --- | --- |
| Academy of Finland | 132611 | Outi Savolainen |
| Academy of Finland | 339702 | Tuomas Hämälä |

The funders had no role in study design, data collection and interpretation, or the decision to submit the work for publication.

### Author contributions

Tuomas Hämälä, Conceptualization, Resources, Data curation, Software, Formal analysis, Funding acquisition, Validation, Investigation, Visualization, Methodology, Writing – original draft; Weixuan Ning, Data curation, Formal analysis, Investigation, Writing – review and editing; Helmi Kuittinen, Conceptualization, Supervision; Nader Aryamanesh, Conceptualization, Resources, Data curation, Formal analysis, Supervision, Investigation, Methodology, Project administration, Writing – review and editing; Outi Savolainen, Conceptualization, Resources, Supervision, Funding acquisition, Methodology, Project administration, Writing – review and editing

### Author ORCIDs

Tuomas Hämälä (iD) http://orcid.org/0000-0001-8306-3397

Decision letter and Author response
Decision letter https://doi.org/10.7554/eLife.83115.sa1
Author response https://doi.org/10.7554/eLife.83115.sa2

## Additional files

### Supplementary files

- Supplementary file 1. Number of read pairs and mapping rates for each sample.
- Supplementary file 2. Bisulfite conversion efficacy for each sample.
- Supplementary file 3. Species used in estimating nucleotide conservation with GERP++.
- MDAR checklist

### Data availability

The transcriptome and whole-genome bisulfite sequencing data are available at NCBI SRA: PRJNA459481. Scripts for conducting the analyses are available at: https://github.com/thamala/lyra-taRnaMet, (copy archived at swh:1:rev:79c3fb242e5acf52d5a1711c739db416a065ef42).

The following dataset was generated:

| Author(s) | Year | Dataset title | Dataset URL | Database and Identifier |
|---|---|---|---|---|
| Hämälä T, Ning W, Kuittinen H, Aryamanesh N, Savolainen O | 2022 | Environmental response in gene expression and DNA methylation reveals factors influencing the adaptive potential of Arabidopsis lyrata | https://www.ncbi.nlm.nih.gov/bioproject/PRJNA459481/ | NCBI BioProject, PRJNA459481 |

The following previously published datasets were used:

| Author(s) | Year | Dataset title | Dataset URL | Database and Identifier |
|---|---|---|---|---|
| Mattila TM, Tyrmi J, Pyhäjärvi T, Savolainen O | 2017 | Genome-wide analysis of colonization history and concomitant selection in Arabidopsis lyrata | https://www.ncbi.nlm.nih.gov/sra/?term=PRJNA357372 | NCBI BioProject, PRJNA357372 |
| Takou M, Hämälä T, Koch EM, Steige KA, Dittberner H, Yant L, Genete M, Sunyaev S, Castric V, Vekemans X, Savolainen O, MJ De | 2021 | Maintenance of adaptive dynamics and no detectable load in a range-edge outcrossing plant population | https://www.ebi.ac.uk/ena/browser/view/PRJEB33206?show=reads | ENA, PRJEB33206 |

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

## Appendix 1

## Quantification of batch effect on detection of differentially expressed genes

Because our samples were sequenced on four lanes, we first determined whether sequencing batches may confound the detection of differentially expressed genes (DEGs). To do so, we examined the results of stably expressed reference genes, which, compared to other genes, are less likely to be influenced by treatment (here field site and population) but equally likely influenced by batch effect. We used the direct *Arabidopsis thaliana* orthologs of 20 genes identified by *Czechowski et al., 2005* and 13 genes identified by *Kudo et al., 2016*. We normalized the read counts with variance stabilization transformation in DESeq2 and used principal component analysis to identify the main sources of variation. Although these reference genes have shown stable expression across treatments in *A. thaliana*, four first principal components from both gene sets were primarily impacted by the field site and population rather than sequencing batch (*Appendix 1—figure 1*). We also found no DEGs (see main text for details) between batches B1 and B2 (containing samples from the high-altitude field) or between batches B3 and B4 (containing samples from the low-altitude field), whereas 4 of the 33 genes were differentially expressed between the field sites. Therefore, the sequencing batches likely have only a minor influence on our results, and by focusing on top-ranking DEGs, we are more likely to capture effects resulting from the growing environment and population history of our experimental plants.

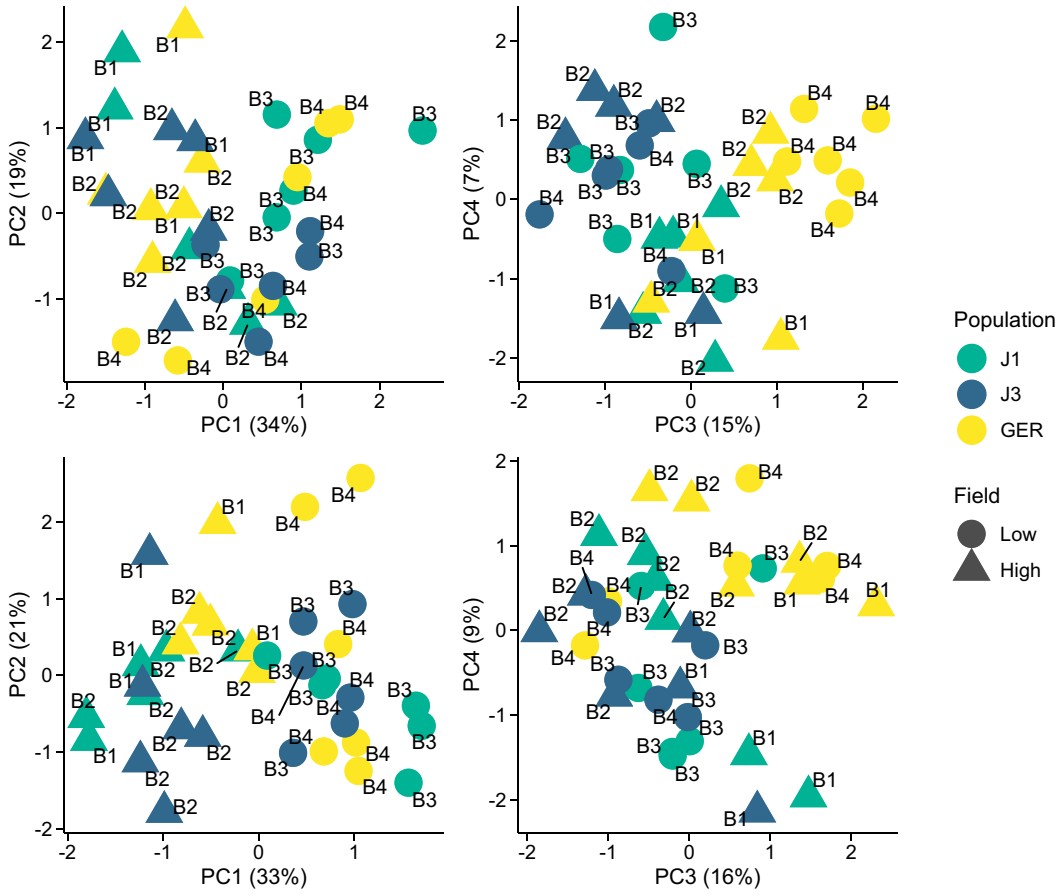

**Appendix 1—figure 1.** Expression variation along first four eigenvectors of a principal component analysis (PCA) conducted using stably expressed reference genes. The sequencing batch (B1, B2, B3, and B4) is marked next to the symbols. Top panels: 20 genes from *Czechowski et al., 2005*. Bottom panels: 13 genes from *Kudo et al., 2016*.

