## [Editor Report]

This work provides a thorough look at changes in expression, methylation, and nucleotide and transposable element diversity among three populations of Arabidopsis lyrata in two different environments. It is a rich dataset, and the authors present a number of nice findings with relevance for our understanding of local adaptation and the process of – and potential constraints to – adaptation to rapid climate change.

---

## [Decision Letter]

**Decision letter after peer review:**

[Editors’ note: the authors submitted for reconsideration following the decision after peer review. What follows is the decision letter after the first round of review.]

Thank you for submitting the paper "Environmental response in gene expression and DNA methylation reveals factors influencing the adaptive potential of *Arabidopsis lyrata*" for consideration by *eLife*. Your article has been reviewed by 3 peer reviewers, including Jeffrey Ross-Ibarra as the Reviewing Editor and Reviewer #1, and the evaluation has been overseen by a Senior Editor.

Comments to the Authors:

Regrettably, after consultation among the reviewers, we have agreed that the work in its current form cannot be considered further for publication by *eLife*. As you will see, there was considerable enthusiasm from the reviewers for what you are trying to achieve, but they also agreed that it is currently unclear whether the analyses are robust enough for the far-reaching inferences made. Having said this, we remain very interested in the work, and would reconsider an extensively revised paper, albeit as a new submission. We would of course try to retain the same reviewers, should you decide to come back to us with such a manuscript.

*Reviewer #1 (Recommendations for the authors):*

I very much liked the author's goal here of going beyond SNP data to evaluate changes in expression, methylation, and TE polymorphism in common garden experiments in order to better understand adaptation.

They really dig into this data and have a number of nice findings.

My main concern is with the overall interpretation of the findings.

The authors claim their TE results show that TEs could be an important source of adaptive variation. While this is certainly true in principle (anything that creates novel genetic variation could be potentially adaptive), I have trouble seeing how the results move us past this? For example, if we interpret the data as showing increased insertion specifically in environmentally responsive genes, and most TEs are deleterious and we see no evidence of TEs generating adaptive change in these data, what have we learned beyond the above? Also, if I'm interpreting correctly, of the 12 DEG-field or DEG-field:population enrichment comparisons with retroelements, enrichment is only found in 3. How does this constitute strong evidence of enrichment?

The authors also claim that evidence of conservation at environmentally responsive genes will constrain adaptation. I have trouble understanding this argument. Does constraint in the past necessarily predict inability to adapt in the future to novel climates? If these are the genes that are most responsive expression-wise, and those expression changes are adaptive, wouldn't these be the genes most likely to see subsequent cis-regulatory change canalize adaptive expression? The total list of DE genes is a few hundred, but if most traits are polygenic, how much would adaptation really be limited even if these genes remain constrained? Finally, what is the interpretation of the strong enrichment of DEG~Field:Pop for number of sweeps? This seems inconsistent with the argument these genes are super conserved and/or constrained for adaptation.

Many of the differences seen in the paper are between German and the Norwegian populations. I would like to see more discussion of how to interpret this, especially in the light of adaptation. We know Germany is poorly adapted to both sites, so what does DE or DM in Germany tell us about adaptation?

Methodological questions:

– How do you deal with multiply mapping reads in evaluating TE expression and methylation?

– Please verify you used a version of RAISd that accounts for variable numbers of bp sequenced in each window.

– To correct bismark, for some genes, WGS SNP data were used, and RNA-seq for others. Do these two sets (WGS genes and RNA-seq genes) show differential methylation?

– There are a number of enrichment analyses done in the paper; are these corrected for multiple testing?

– Figure 6D looks at the % of TEs. Are total numbers of new DNA insertions higher in J3 than J1? That would change the interpretation I think?

– Does Figure 4 look different if you plot the 95% range rather than the CI?

– It might be useful to quantify TE expression and methylation following Anderson (https://pubmed.ncbi.nlm.nih.gov/31506319/).

– I believe newer versions of RIASd can account for variable bp per window. Since you have this number from ANGSD, you can rescale the RAISd mu statistic appropriately if not.

– I might move Figure 3 to supplement. though good to document, I didn't find it super helpful.

*Reviewer #2 (Recommendations for the authors):*

This manuscript reports on a genomic analysis of an intricate common garden experiment utilizing three populations (two local and one distant) to better understand the evolution or adaptive benefit of gene expression variation. This is a very interesting paper but there was a bit of an assumption about how DMGs and DEGs are cis variants that needs strengthening or supporting. Without this support, it isn't clear if the evolution of cis sequences is informative of what is happening with DMGs if they are largely trans effects as one might see for a FRI/FLC or similar polymorphism causing large transcriptomic effects.

One complication I'm struggling with is the analysis of differentially expressed genes. The analysis directly compares DEGs to DMGs and to genetic variation with an implicit assumption that they are comparable. However, DEGs can be caused by cis or trans events such that the majority of DEGs are caused by a single causal locus. This maybe comes up the strongest when the analysis shifts to looking for footprints of selection at the DEGs (line 225). This seems to assume the DEGs are all caused by cis but it isn't clear what the support is for this claim. Is there evidence that these are mainly cis (e.g. loci with known PAVs, large bimodal effects, etc)? Some support should be provided for an assumption that a DEG is being caused in cis.

In contrast, DMGs are almost always cis as shown by work from the Springer and Schmitz groups. Although there are known loci influencing global methylation levels which does appear to be the case with the German collection having lower methylation across the board as well as a lack of relative plasticity (Figure 2A and 2D) in comparison. How does this lack of plasticity influence the ability to assess the DMG ~ Field and DMG ~ Field:Population terms as the overall methylation in the German collection is non plastic. In the local populations, one could assume the DMGs are local events but when comparing J1/J2 to Ger there is the possibility that DMGs are a blend of cis/trans. Is there evidence similarly supporting DMGs in this comparison as mainly cis?

A similar concern is that in the closely related *Arabidopsis thaliana*, a large fraction of cis causality for DEGs is actually not promoter polymorphisms but actually PAV or other structural variation within the loci. Is there evidence in this collection that cis causality is mainly SNP based as the analysis on selection is suggesting? This is partly a concern because, at least in *A. thaliana*, this can lead to these loci having an elevated level of error in short read sequencing leading to a ponderance of rare SNPs (See recent work by Nordborg and colleagues on variation in gene duplication and SNP error rates).

Similarly work by Schmitz and Ecker has shown that DMGs can also be biased towards genes with structural variants. In this studies, DEGs and DMGs seem to not be behaving in coordination so structural variants aren't likely a universal problem but some assessment of their potential in influencing the data would be necessary to understand how pangenome variation may be introducing error.

*Reviewer #3 (Recommendations for the authors):*

This study was aimed at identifying the adaptive potential of genomes in response to different environments. To accomplish this aim, the authors performed reciprocal transplant experiments with distinct populations of Arabidopsis lyrata. These plants were grown over a year and then tissue was harvested and used for gene expression and DNA methylation profiling genome wide. A strength of this research is the unique samples available to investigate. This allows the evaluation of impact on differential expression/methylation to be attributed to the environment, genotype or both. Many of the conclusions are justified, but analysis of the DNA methylation data is confounded by the way it was defined and analyzed. The weakness in this analysis stems from only investigating the impact of CG methylation. It is well established in plant genomes (including A. lyrata) that there are two distinct categories of methylated regions. One region is defined as GBM (gene body DNA methylation), which is defined by CG-only methylation in a gene body. The second category of methylation is found anywhere in the genome (including genes) and contains CG + non-CG (CHG and CHH) methylation. This study has not properly distinguished these regions and it has likely led to improper conclusions.

With regards to the findings presented on plasticity of DNA methylation, many of the findings are somewhat expected. It has been established that DNA methylation variation is largely driven by genetic variation within plant genomes. The variation in methylation found between different environments on non-gene regions also reflects common observations. There are lots of environments that lead to genome-wide changes in methylation. It's important to note that not all of these changes are causal. In fact, many of them are likely an indirect effect of 3D nuclear genome compaction and access of DNA methyltransferases to target sequences. Notice the changes in methylation are very subtle and they exclusively occur at regions that already possess methylation. It is not like mammalian systems where methylation is completely lost or gained at regions.

The section of TE activation by stress was not fully supported by the data presented. It's an intriguing possibility, but not well supported by the data. This entire section needs to be revisited with additional analyses.

Overall, although this is a unique population the conclusions presented are incremental to the field. The shortcomings presented in the analysis can be easily remedied, although it will be curious to see how they change the conclusions.

The DNA methylation data must be partitioned into CG and CG/CHG/CHH regions in the analysis. Using CG methylation alone mixes two distinct groups together. It is well established that CG/CHG/CHH methylated regions are generally silenced whereas CG only methylated genes are expressed.

More information is required for understanding how DMGs were defined. What is the size distribution of DMGs? What proportion of DMGs are GBM vs those that contain CG/CHG/CHH? The rapidly evolving DMGs mentioned on line 252 are likely those that contain CG/CHGCHH, as worked by Gaut et al. have shown that GBM genes have low rates of nucleotide substitutions.

I'm not convinced by the TE analysis. Are these full length TEs? Or are they TE fragments? How are multiple mapping reads handled in the RNA-seq analysis? Read counts of 5-6 as presented in 6b are not very convincing for "expressed" TEs. I found this to be another weak aspect of the study. What's the evidence these TEs are "stress induced"? It is stated that TEs overlapping genes were removed, but these genotypes were mapped to a reference genome? There will be differences and no genomes have 100% annotation. This is especially true of out-crossing species. Overall, I found the evidence of stress induced activation of TEs not well supported by the data.

Line 405 – there is not assignment of cause and effect from the results of this study. The authors should be careful not to state methylation is affecting expression. It could equally be possible that expression is affecting methylation.

I found the section on rates of evolution of 1kb upstream regions worthwhile, but to extend this make conclusions about cis-regulatory elements is a bit of a stretch. No CREs were identified and there are lots of non-CREs located upstream of genes that could impact these results.

The bisulfite conversion rates need to be presented for each of the 24 samples instead of an average.

The number of reads sequenced per RNA-seq/BS-seq library need to be presented in a supplementary table along with their alignments rates (and bs conversion rates).

---

## [Author Response]

[Editors’ note: the authors resubmitted a revised version of the paper for consideration. What follows is the authors’ response to the first round of review.]

Reviewer #1 (Recommendations for the authors):I very much liked the author's goal here of going beyond SNP data to evaluate changes in expression, methylation, and TE polymorphism in common garden experiments in order to better understand adaptation.They really dig into this data and have a number of nice findings.My main concern is with the overall interpretation of the findings.The authors also claim that evidence of conservation at environmentally responsive genes will constrain adaptation. I have trouble understanding this argument. Does constraint in the past necessarily predict inability to adapt in the future to novel climates? If these are the genes that are most responsive expression-wise, and those expression changes are adaptive, wouldn't these be the genes most likely to see subsequent cis-regulatory change canalize adaptive expression? The total list of DE genes is a few hundred, but if most traits are polygenic, how much would adaptation really be limited even if these genes remain constrained? Finally, what is the interpretation of the strong enrichment of DEG~Field:Pop for number of sweeps? This seems inconsistent with the argument these genes are super conserved and/or constrained for adaptation.

We agree that plastic expression responses are more likely to be canalized at adaptive genes. However, the idea of canalization (or genetic assimilation) relies on heritable variation existing for plasticity, so that selection can act on such variation in new selective environments. Our reasoning here is that strong selective constraint at both the coding and regulatory regions of environmentally responsive genes keeps the pool of segregating variants small, and any subsequent adaptive variants need to arise de novo. This would likely decrease the rate of adaptive evolution compared to a case where variants are readily available, especially under rapid environmental change. We now make this point clearer in the manuscript (lines 407 – 411).

The question about polygenicity is a tough one, as we don’t know how polygenic the adaptive traits are. We may assume that the genetic architecture is highly polygenic, and that the loci identified here are primarily of large effect (although, we would argue that 3933 environmentally responsive genes, which is 18% of all expressed genes, is already a nontrivial number). In that case, high polygenicity would reduce the effectiveness of directional selection (compared to an oligogenic architecture), and if also the loci with largest effect are constrained, this could have a considerable influence on the adaptive potential of these populations. We now mention this point of the effect sizes at line 189. However, as similar questions could be asked of almost any genomicsoriented local adaptation studies (i.e., how much do selection outliers / GWAS hits actually contribute to adaptation), we keep this section rather brief.

Last, increased number of sweeps in DEG ~ Field:Population genes might, at first glance, seem inconsistent with the strong signals of selective constraint at coding regions. However, as we now more carefully describe (line 256), sweep signals were primarily found upstream of the TSS. We also detect low diversity and high differentiation at these regions, suggesting positive selection may have acted on cisregulatory regions. Given that we already found genetically variable expression responses at these genes, less constrained regulatory variants are perhaps not so surprising.

The authors claim their TE results show that TEs could be an important source of adaptive variation. While this is certainly true in principle (anything that creates novel genetic variation could be potentially adaptive), I have trouble seeing how the results move us past this? For example, if we interpret the data as showing increased insertion specifically in environmentally responsive genes, and most TEs are deleterious and we see no evidence of TEs generating adaptive change in these data, what have we learned beyond the above? Also, if I'm interpreting correctly, of the 12 DEG-field or DEG-field:population enrichment comparisons with retroelements, enrichment is only found in 3. How does this constitute strong evidence of enrichment?

It is true that out of the six different plastic gene groups we find strong TE enrichment in only one of them (DEG ~ Field:Population in J3), but we think that this is consistent with the expression and methylation results. We find higher TE expression and methylation levels at the high-altitude field site, indicating stress-induced TE activation. We see the same pattern in all three populations (although the signal is more subtle in GER). However, out of the whole-genome sequenced individuals, only J3 population is native to the alpine environment and therefore could have accumulated recent TE insertions due to the activation. We don’t see an enrichment of TEs in the DEG ~ Field genes in J3, which could be related to them being more constrained than DEG ~ Field:Population genes (lines 439 – 441). Therefore, these results, combined with previous results from *A. thaliana*, suggest that certain LTRs may be preferentially integrating into the gene bodies of environmentally responsive genes, but are still being purged from the most highly constrained genes.

Our reasoning related to the potentially adaptive effects of TEs stems from the points made above, namely that canalization of plastic effects relies on existing genetic variation: these genes have a smaller pool of segregating SNPs but a larger pool of segregating TEs (on top of that, they may be more likely targeted by new TE insertions, which can happen at a considerably higher rate than point mutations). Therefore, there is an increased chance that any functional effects will arise as a result of TE insertions, although we acknowledge that most such insertions are likely detrimental. We realize that some of this discussion is quite speculative, and we have now moved it to the new “Ideas and speculation” section.

Many of the differences seen in the paper are between German and the Norwegian populations. I would like to see more discussion of how to interpret this, especially in the light of adaptation. We know Germany is poorly adapted to both sites, so what does DE or DM in Germany tell us about adaptation?

We have added more discussion about this topic in the manuscript (lines 358 – 365). In short, although much of the expression (and potentially methylation) differentiation in GER is likely maladaptive, many differences likely reflect evolved responses in the Norwegian populations (i.e., the GER response is ancestral). Furthermore, as most expression responses were consistent in the three populations, including GER informs about the conservation of the environmental effects.

Methodological questions:– How do you deal with multiply mapping reads in evaluating TE expression and methylation?

Earlier we only used uniquely mapped reads, but now we also included multi mapping reads in the quantification of TE expression (following Anderson et al. 2019). Methylation calling still only uses uniquely mapped reads because it is done site-by-site basis (i.e., the same way as SNP calling) and we are not aware of an approach that accurately takes multimapping reads into account.

– Please verify you used a version of RAISd that accounts for variable numbers of bp sequenced in each window.

Thanks for pointing this out. We used the newest version of RAiSD, but it appears that the unmappable regions need to be excluded using the -X option, which we didn’t do before. We now removed regions that contained no sequencing data.

– To correct bismark, for some genes, WGS SNP data were used, and RNA-seq for others. Do these two sets (WGS genes and RNA-seq genes) show differential methylation?

Genes not found in the RNA-seq data do have higher non-CG methylation levels than expressed genes (CG_expressed_ = 0.270, CHG_expressed_ = 0.0156, CHH_expressed_ = 0.006; CG_non-expresse_d = 0.231, CHG _non-expressed_ = 0.151, CHH _non-expressed_ = 0.038). However, this pattern is expected, because higher non-CG methylation is associated with lower gene expression (line 190), and it is unlikely that the difference would be an artifact of the correction (if indeed this is what the reviewer had in mind).

– There are a number of enrichment analyses done in the paper; are these corrected for multiple testing?

Yes. We have now been more careful to add Q values to all enrichment tests.

– Figure 6D looks at the % of TEs. Are total numbers of new DNA insertions higher in J3 than J1? That would change the interpretation I think?

We focus on frequencies and percentages because the total number of TE insertions is strongly dependent on the samples, with higher sequencing coverage resulting in more TEs. It is true that if the total number of DNA transposons is higher in J3 than in J1, then more efficient purging in J3 would be unlikely. In fact, it may be equally likely that both class I and II TEs are more numerous in J3, but we see relatively more class I TEs simply because they move through copying (i.e., it has nothing to do with purging). We have modified the text to take this more careful interpretation into account (lines 314 – 320).

– Does Figure 4 look different if you plot the 95% range rather than the CI?

Ranges do overlap between the different candidate gene groups, but we don’t think that this is inconsistent with any of our interpretation of the results. Among hundreds/thousands of genes, one would not expect that nearly all are non-overlapping.

– It might be useful to quantify TE expression and methylation following Anderson (https://pubmed.ncbi.nlm.nih.gov/31506319/).

Thanks for the suggestion. We have now quantified TE expression using a similar approach as Anderson et al. (although with slightly different methods). However, as estimation of methylation levels requires site-level information, it is best done using uniquely mapped reads.

– I believe newer versions of RIASd can account for variable bp per window. Since you have this number from ANGSD, you can rescale the RAISd mu statistic appropriately if not.

Thanks for pointing this out. As we mention above, we now excluded regions that contained no sequencing data.

– I might move Figure 3 to supplement. though good to document, I didn't find it super helpful.

That’s a good point. That figure is now part of the supplement.

Reviewer #2 (Recommendations for the authors):This manuscript reports on a genomic analysis of an intricate common garden experiment utilizing three populations (two local and one distant) to better understand the evolution or adaptive benefit of gene expression variation. This is a very interesting paper but there was a bit of an assumption about how DMGs and DEGs are cis variants that needs strengthening or supporting. Without this support, it isn't clear if the evolution of cis sequences is informative of what is happening with DMGs if they are largely trans effects as one might see for a FRI/FLC or similar polymorphism causing large transcriptomic effects.One complication I'm struggling with is the analysis of differentially expressed genes. The analysis directly compares DEGs to DMGs and to genetic variation with an implicit assumption that they are comparable. However, DEGs can be caused by cis or trans events such that the majority of DEGs are caused by a single causal locus. This maybe comes up the strongest when the analysis shifts to looking for footprints of selection at the DEGs (line 225). This seems to assume the DEGs are all caused by cis but it isn't clear what the support is for this claim. Is there evidence that these are mainly cis (e.g. loci with known PAVs, large bimodal effects, etc)? Some support should be provided for an assumption that a DEG is being caused in cis.

Thanks for the insightful comments. We have now been more careful in interpreting our results and described our reasoning in more detail in the manuscript. That said, we don’t actually assume that the observed differences are only due cis variants: we examined putative promoter regions because we were able to do so using the current dataset. In the manuscript we study two processes that may influence adaptive phenotypes: variation in expression levels, which is primarily affected by cis and trans regulatory elements, and variation in the gene product, which is primarily affected by the coding sequence. Our first observation is that the environmentally induced expression responses are highly consistent across the three populations (and any differences are mainly between Norway and Germany), indicating that the regulatory systems are largely conserved. For cis elements, this assumption was supported by our analysis of the promoter regions. Although we can’t conduct similar analysis on trans-regulatory elements, we believe them to be similarly conserved (line 388). We then find that the coding sequences (and by extension, the gene products) of the environmentally responsive genes are highly constrained. Therefore, given the lack of heritable variation at the expression and sequence levels, we infer that the environmentally responsive genes have weak potential for evolutionary change.

In contrast, DMGs are almost always cis as shown by work from the Springer and Schmitz groups. Although there are known loci influencing global methylation levels which does appear to be the case with the German collection having lower methylation across the board as well as a lack of relative plasticity (Figure 2A and 2D) in comparison. How does this lack of plasticity influence the ability to assess the DMG ~ Field and DMG ~ Field:Population terms as the overall methylation in the German collection is non plastic. In the local populations, one could assume the DMGs are local events but when comparing J1/J2 to Ger there is the possibility that DMGs are a blend of cis/trans. Is there evidence similarly supporting DMGs in this comparison as mainly cis?

It is true that overall methylation in GER was less plastic that in the Norwegian populations, but we don’t think that this has a major influence on our ability identify to detect DMGs (or at least we correct for this as much as possible). For finding DMGs due to field site, we compare the fit of models: methylation ~ population + field and methylation ~ population (we now describe the models in Table S1). Since GER has fewer plastic effects and lower methylation across the board, the population term attempts to correct for this effect. Of course, the lower methylation levels and more subtle plastic effects likely reduce the power to detect DMGs, but since our goal was to find representative gene sets for quantification of selective constraint, not to identify all DMGs, we believe our conclusion to be valid.

A similar concern is that in the closely related Arabidopsis thaliana, a large fraction of cis causality for DEGs is actually not promoter polymorphisms but actually PAV or other structural variation within the loci. Is there evidence in this collection that cis causality is mainly SNP based as the analysis on selection is suggesting? This is partly a concern because, at least in *A. thaliana*, this can lead to these loci having an elevated level of error in short read sequencing leading to a ponderance of rare SNPs (See recent work by Nordborg and colleagues on variation in gene duplication and SNP error rates).Similarly work by Schmitz and Ecker has shown that DMGs can also be biased towards genes with structural variants. In this studies, DEGs and DMGs seem to not be behaving in coordination so structural variants aren't likely a universal problem but some assessment of their potential in influencing the data would be necessary to understand how pangenome variation may be introducing error.

It is possible that SVs may underlie some of the genetically based expression differences observed here (note that most of the environmental effects were plastic and thus unlikely to be influenced by SVs). For example, we found increased Fst within the promoter regions of DEG ~ Field:Population genes, which could, at least in part, be due to SVs that are in LD with the SNPs. However, given that these types of analyses don’t assume that any of the SNPs are causative (just that they are in LD with the causative variant), such a scenario would not invalidate our results. Furthermore, we think that a comparison to *A. thaliana* is not as straight forward as one might think: large-effect SVs are well known to influence flowering time in *A. thaliana* (as the reviewer points out), but no evidence of similar effects have been found in A. lyrata (Kuittinen et al. 2008). This difference could be related to the mating system, as large effect loci may be less likely to reach high frequencies in outcrossing species (as also found in maize, Buckler et al. 2009).

We agree that SVs within the coding regions would complicate the alignment of sequencing reads, but for plastic DEGs this seems like an unlikely source of error. First, for this to cause the observed expression difference, SVs would need to be present only in individuals planted either in the low- or high-altitude field site (for few genes this could be the case, but unlikely for the majority). Second, we conducted the selection analysis using independent whole-genome sequenced individuals, meaning that the same SVs could not cause both the expression difference and signals of strong selective constraint. Third, much of our selection analysis is based on comparing the synonymous and nonsynonymous sites (DFE and α), which indicated stronger than expected constraint at functional sites. We would not expect alignment errors to primarily influence nonsynonymous sites (indeed, assembly-based studies have found that genes with signals of strong selective constraint are, on average, less likely to overlap SVs, Hämälä et al. 2021). Fourth, our selection inferences were confirmed by two comparative tests (one for coding and for promotor sequence), which are unaffected by SV polymorphisms. In contrast to DEGs, non-CG DMGs may be more likely affected by SVs given the signals of relaxed selective constraint. In fact, we find an enrichment of TE polymorphism at these genes, which could (at least in part) be driving the plastic methylation responses observed at these genes (i.e., TEs are environmentally responsive and methylation spreads to the genic regions).

Reviewer #3 (Recommendations for the authors):This study was aimed at identifying the adaptive potential of genomes in response to different environments. To accomplish this aim, the authors performed reciprocal transplant experiments with distinct populations of Arabidopsis lyrata. These plants were grown over a year and then tissue was harvested and used for gene expression and DNA methylation profiling genome wide. A strength of this research is the unique samples available to investigate. This allows the evaluation of impact on differential expression/methylation to be attributed to the environment, genotype or both. Many of the conclusions are justified, but analysis of the DNA methylation data is confounded by the way it was defined and analyzed. The weakness in this analysis stems from only investigating the impact of CG methylation. It is well established in plant genomes (including A. lyrata) that there are two distinct categories of methylated regions. One region is defined as GBM (gene body DNA methylation), which is defined by CG-only methylation in a gene body. The second category of methylation is found anywhere in the genome (including genes) and contains CG + non-CG (CHG and CHH) methylation. This study has not properly distinguished these regions and it has likely led to improper conclusions.With regards to the findings presented on plasticity of DNA methylation, many of the findings are somewhat expected. It has been established that DNA methylation variation is largely driven by genetic variation within plant genomes. The variation in methylation found between different environments on non-gene regions also reflects common observations. There are lots of environments that lead to genome-wide changes in methylation. It's important to note that not all of these changes are causal. In fact, many of them are likely an indirect effect of 3D nuclear genome compaction and access of DNA methyltransferases to target sequences. Notice the changes in methylation are very subtle and they exclusively occur at regions that already possess methylation. It is not like mammalian systems where methylation is completely lost or gained at regions.The section of TE activation by stress was not fully supported by the data presented. It's an intriguing possibility, but not well supported by the data. This entire section needs to be revisited with additional analyses.Overall, although this is a unique population the conclusions presented are incremental to the field. The shortcomings presented in the analysis can be easily remedied, although it will be curious to see how they change the conclusions.

Thank you for the insightful comments. It does appear that we should have also considered CHG and CHH methylation in our DMG analysis. To remedy this, we now conducted the DMG analysis separately for CG (while controlling for variation in non-CG sites) and non-CG methylation. As pointed out by the reviewer, our previous results related to DMGs were likely driven by non-CG methylation (or the combined CG+CHG+CHH methylation). Our revised results show that CG-only methylation at gene bodies shows few environmental effects, consistent with the idea that GBM is highly conserved.

We agree that results related to the plasticity of DNA methylation are not novel, and we cite many previous studies done in *A. thaliana*. However, the examination of methylation variation at genes and TEs is only a small part of our study. The main point was to combine the analysis of plastic and genetic variation in gene expression and DNA methylation with population genetic analyses of selection to assess how such variation influences adaptive evolution. This, to our knowledge, has not been done before.

The DNA methylation data must be partitioned into CG and CG/CHG/CHH regions in the analysis. Using CG methylation alone mixes two distinct groups together. It is well established that CG/CHG/CHH methylated regions are generally silenced whereas CG only methylated genes are expressed.More information is required for understanding how DMGs were defined. What is the size distribution of DMGs? What proportion of DMGs are GBM vs those that contain CG/CHG/CHH? The rapidly evolving DMGs mentioned on line 252 are likely those that contain CG/CHGCHH, as worked by Gaut et al. have shown that GBM genes have low rates of nucleotide substitutions.

Thank you for the suggestion. As we describe above, we now separate CG and non-CG DMGs and find that most methylation changes occur in the non-CG sites (consistent with the idea that GBM is highly conserved). We define DMGs as genes with differential methylation in the gene bodies and therefore sizes of the DMGs depend on the length of the genes.

I'm not convinced by the TE analysis. Are these full length TEs? Or are they TE fragments? How are multiple mapping reads handled in the RNA-seq analysis? Read counts of 5-6 as presented in 6b are not very convincing for "expressed" TEs. I found this to be another weak aspect of the study. What's the evidence these TEs are "stress induced"? It is stated that TEs overlapping genes were removed, but these genotypes were mapped to a reference genome? There will be differences and no genomes have 100% annotation. This is especially true of out-crossing species. Overall, I found the evidence of stress induced activation of TEs not well supported by the data.

The consensus sequences are full length TEs but reads may also map to TE fragments. To strengthen our TE analysis, we followed reviewer 1’s suggestion and quantified TE expression using the same approach as Anderson et al. 2019 (quantification at the level of TE families, using also multimapping reads). Although we would expect that only full length TEs are expressed, reads may also map to TE fragments due to sequence similarity. The approach we use now takes those reads better into account by assigning them to TE families.

We have now added a unit to the y-axis label of 5B: what’s shown are read counts after variance stabilization transformation (to account for different library sizes), not raw counts (which have an average of around 80 reads). Subsequently, we observed considerably higher TE expression levels at the more stressful high-altitude field site. This result may be due to three reasons: stress induced TE activation, technical bias (e.g., batch effect), or unaccounted gene expression. We have now conducted additional analysis to rule out technical bias and gene expression (line 293). Briefly, technical bias would influence the whole library, whereas we only see higher TE (and not gene) expression in the high-altitude field (Figure S14). Similar reasoning also applies to the effects of genes that have been missed in the genome annotation. It is unlikely that such missing genes would be systematically upregulated in the high-altitude field site, especially given that differentially expressed genes exhibited no such bias.

Line 405 – there is not assignment of cause and effect from the results of this study. The authors should be careful not to state methylation is affecting expression. It could equally be possible that expression is affecting methylation.

That’s a fair point. We have modified the sentence (line 384).

I found the section on rates of evolution of 1kb upstream regions worthwhile, but to extend this make conclusions about cis-regulatory elements is a bit of a stretch. No CREs were identified and there are lots of non-CREs located upstream of genes that could impact these results.

We have now been more careful in describing these results and acknowledge that selection may have also acted on other parts of the transcriptional machinery (line 220).

The bisulfite conversion rates need to be presented for each of the 24 samples instead of an average.The number of reads sequenced per RNA-seq/BS-seq library need to be presented in a supplementary table along with their alignments rates (and bs conversion rates).

These are now shown in Tables S2 and S3.